# Aerosol-boundary-layer-monsoon interactions amplify semi-direct effect of biomass smoke on low cloud formation in Southeast Asia

Ke Ding [1,2,3,14], Xin Huang [1,2,3,14], Aijun Ding [1,2,3✉], Minghuai Wang [1,2], Hang Su [4], Veli-Matti Kerminen[5], Tuukka Petäjä [1,5], Zhemin Tan [1,2], Zilin Wang [1], Derong Zhou[1,2], Jianning Sun[1,2], Hong Liao [6], Huijun Wang[7], Ken Carslaw [8], Robert Wood[9], Paquita Zuidema [10], Daniel Rosenfeld[1,11], Markku Kulmala[1,5], Congbin Fu[1,2], Ulrich Pöschl[4], Yafang Cheng [4✉] & Meinrat O. Andreae [4,12,13]

Low clouds play a key role in the Earth-atmosphere energy balance and influence agricultural production and solar-power generation. Smoke aloft has been found to enhance marine stratocumulus through aerosol-cloud interactions, but its role in regions with strong human activities and complex monsoon circulation remains unclear. Here we show that biomass burning aerosols aloft strongly increase the low cloud coverage over both land and ocean in subtropical southeastern Asia. The degree of this enhancement and its spatial extent are comparable to that in the Southeast Atlantic, even though the total biomass burning emissions in Southeast Asia are only one-fifth of those in Southern Africa. We find that a synergetic effect of aerosol-cloud-boundary layer interaction with the monsoon is the main reason for the strong semi-direct effect and enhanced low cloud formation in southeastern Asia.

[1] Joint International Research Laboratory of Atmospheric and Earth System Sciences, School of Atmospheric Sciences, Nanjing University, Nanjing 210023, China. [2] Jiangsu Provincial Collaborative Innovation Center of Climate Change, Nanjing, China. [3] Frontiers Science Center for Critical Earth Material Cycling, Nanjing University, Nanjing 210023, China. [4] Max Planck Institute for Chemistry, Mainz, Germany. [5] Institute for Atmospheric and Earth System Research (INAR)/Physics, University of Helsinki, Helsinki, Finland. [6] Jiangsu Key Laboratory of Atmospheric Environment Monitoring and Pollution Control, Jiangsu Collaborative Innovation Center of Atmospheric Environment and Equipment Technology, School of Environmental Science and Engineering, Nanjing University of Information Science & Technology, Nanjing 210044, China. [7] School of Atmospheric Sciences, Nanjing University of Information and Science Technology, Nanjing 210044, China. [8] Institute for Climate and Atmospheric Science, School of Earth and Environment, University of Leeds, Leeds, UK. [9] Department of Atmospheric Sciences, University of Washington, Seattle, USA. [10] Rosenstiel School of Marine and Atmospheric Sciences, University of Miami, Miami, FL, USA. [11] Institute of Earth Sciences, The Hebrew University of Jerusalem, Jerusalem 91904, Israel. [12] Scripps Institution of Oceanography, University of California San Diego, La Jolla, CA, USA. [13] Department of Geology and Geophysics, King Saud University, Riyadh, Saudi Arabia. [14] These authors contributed equally: Ke Ding, Xin Huang. ✉email: dingaj@nju.edu.cn; yafang.cheng@mpic.de

Low clouds, including stratocumulus, cumulus, and stratus, cover about 30% of the globe and play important roles in the Earth system because of their strong influence on the planetary albedo and the energy balance of the Earth[1,2]. By reducing solar radiation reaching the surface, persistent low clouds over land have negative impacts on agricultural production and solar-power generation[3,4]. Therefore, understanding the factors governing low cloud cover is not only critical for regional weather forecasting and global climate prediction but also important for their socioeconomic effects[3,5–10].

Modeling studies have suggested that light-absorbing aerosols from combustion sources like biomass burning (BB) can enhance the formation and evolution of low clouds, especially over the Southeast Atlantic and adjacent parts of Africa (Atlantic-Africa) in austral winter and spring[11–15]. However, in East Asia, a region that combines high population density, high social and economic relevance, high levels of air pollution driving aerosol–climate interactions[16,17], and strong variability of monsoon circulations[18–21], a quantitative understanding of these effects on low clouds is still missing, especially from a climatological perspective[22–24].

By analyzing 16 years of satellite observations and meteorological reanalysis data together with numerical model simulations, we find that the efficiency of low-cloud enhancement by the smoke aloft is particularly strong in Asia, with the low cloud fraction increasing by a factor of four more than that in the Southeast Atlantic. The amplified aerosol–cloud interactions are mainly caused by the smoke's coupling with planetary boundary layer (PBL) and the Asian winter monsoon.

## Results

### Evidence from observation minus reanalysis (OMR) analysis.
We adopted an OMR approach, combining MODIS satellite observational data with ECMWF (European Centre for Medium-Range Weather Forecast) Interim Re-Analysis (ERA-Interim) data. Since all observations used in ERA-Interim are subject to quality control and data selection, observations whose departures from the model priors exceed prescribed thresholds are not assimilated or have no influence on the analyses[25–27]. Therefore, the OMR approach can shed light on the aerosols' effects that have not been included in the reanalysis[27–30].

Figure 1a, b shows the 16-year averaged OMR difference in cloud fraction during the BB seasons in southeastern Asia (March) and Atlantic-Africa (August), two regions of the globe with intensive BB and cloud-induced outgoing short-wave radiation (Supplementary Figs. 1 and 2). The high OMR values are mostly distributed along the coastal region and the oceanic area off southern China and the Atlantic off southwestern Africa, covering areas over $3000 \times 1000$ km$^2$ and featuring mostly low clouds (Supplementary Fig. 1b). Although the total BB emission in southeastern Asia in March–April, predominately from forest and agricultural fires in the northern Indochina Peninsula, is only 20% of that in southern Africa in June–August (Supplementary Table 1), the cloud cover in the main transport pathways of the smoke shows a similar enhancement (over 30%) in both cases (Fig. 1a, b), suggesting a much stronger aerosol effect on low cloud formation in southeastern Asia.

The stronger aerosol effect in southeastern Asia is not only evident from the climatological average but also resolved at different tempo-spatial scales. As shown in Fig. 1c and Supplementary Fig. 3, the inter-annual variability of OMRs follows that of the aerosol optical depth (AOD), BB emissions, and CO column density in the middle troposphere, with a better correlation in Asia than in Africa. Such positive correlations also hold for the monthly and daily averaged data (Fig. 1d). As demonstrated in Fig. 2, the averaged vertical distribution of CALIPSO aerosol extinction and cloud occurrence for high and low-AOD years also suggest a stronger vertical linkage of smoke

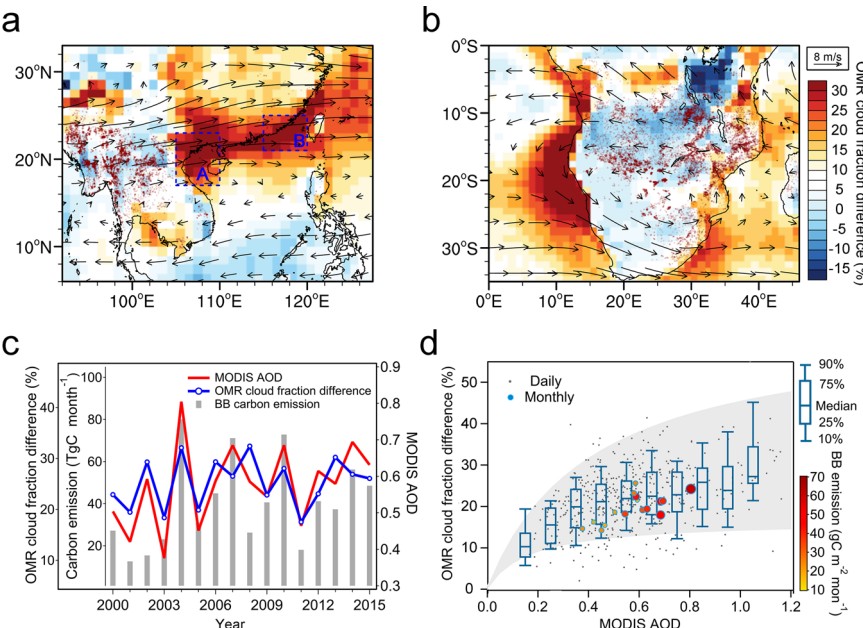

**Fig. 1 Observation minus reanalysis (OMR) difference in cloud fraction and its relationship to aerosol optical depth (AOD). a, b** OMR differences in cloud fraction (with MODIS cloud amount as the observation) together with 700 hPa wind fields in March in Asia and August in Africa, respectively, during 2000–2015. **c** Time series of monthly averaged biomass-burning (BB) carbon emission, OMR difference in cloud fraction (with MODIS cloud amount as the observation), and MODIS AOD for Asia in March (the region for averaging is denoted in Supplementary Fig. 3a). **d** Cloud enhancement as a function of AOD in subtropical southeastern Asia in March during 2000–2015. Note: The dark red dots in (**a**) and (**b**) show satellite-detected fire counts. The blue boxes labeled A and B in (**a**) define regions with the highest OMR values for further analysis in Fig. 2. In (**d**), gray dots show daily regional results and the whisker-box plot gives the statistics of daily data with different AOD bins. The daytime cloud enhancement between the 10th and 90th percentiles is marked as the gray shading, and monthly results are color-coded with marker size indicating biomass-burning emission in the source region.

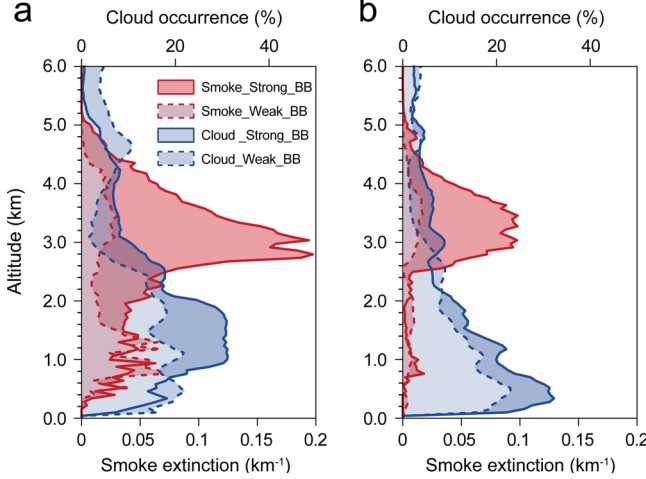

**Fig. 2 Relationship between smoke and cloud occurrences measured by the CALIPSO satellite instruments.** Averaged vertical profiles of Cloud-Aerosol Lidar and Infrared Pathfinder Satellite Observation (CALIPSO) smoke extinction and cloud occurrence in the 3 years with the highest (indicated as "Strong_BB") and the lowest (indicated as "Weak_BB") smoke aloft during 2007–2015 for **a** Baibu Bay (2010, 2012, 2014 as high years compared to 2008, 2009, 2011) and **b** the Taiwan Strait (2007, 2010, 2015 as high years compared to 2008, 2012, 2013). Region definitions are given in Fig. 1a. The highest and the lowest 3 years were classified according to the column smoke extinction between 2 and 5 km, excluding years with inconsistent aerosol optical depth and extremely high smoke aerosol concentrations above the cloud.

aloft and low-cloud enhancement below in Asia than in the Atlantic-Africa region (Supplementary Figs. 4 and 5). In Asia, the low-cloud enhancement mainly concentrates beneath the BB plume around 3 km[31–34], with cloud top heights of approximately 2.5 km above sea level over the lee side of the Yungui and Shan plateaus (i.e., the Beibu Bay, Box A in Fig. 1a) (Fig. 2a) and cloud top heights of approximately 1.5 km above the flatlands and ocean (i.e., the Taiwan Strait, Box B in Fig. 1a) (Fig. 2b) (the geographical definitions are given in Supplementary Fig. 6).

Over the Beibu Bay, a strong increase in cloud occurrence (50–60%) exists between the altitudes of 1–2 km in high BB pollution years (Fig. 2a). In this region, the low-cloud enhancement exists not only over ocean but also over land areas (Fig. 1a), which is also one of the distinctive features in subtropical Asia in contrast to Atlantic-Africa. In the latter region (Fig. 1b), low clouds are enhanced over the Atlantic but reduced on the African continent, with coastal lines as a clear border between the two distinct effects, which have been well documented in previous modeling studies and satellite observations[5,13,14,35–37]. The enhancement over the eastern Atlantic Ocean has been attributed to weakened cloud-top entrainment of overlying dry air due to absorbing aerosols from BB above the marine stratocumulus[12,38], possibly aided by BB aerosol interacting with cloud layer[39,40]. Further to the northwest, the cloud reduction has been observationally linked to an aerosol-induced increase in temperature reducing the relative humidty[41]. In contrast, the reduction of cloudiness over land in Africa can be explained by the aerosol absorption cloud fraction feedback (AFF) proposed by Koren et al.[42], where surface cooling reduces moisture fluxes and the upper-level warming reduces the relative humidity in the cloud layer. This raises the questions: What causes the different response over the land regions between Africa and Asia, and how is this connected to the mechanisms of low-cloud amplification in subtropical Asia?

**Mechanistic understanding based on numerical modeling**. To understand these underlying mechanisms, we performed numerical simulations with the chemistry–meteorology online-coupled WRF-Chem model (Weather Research and Forecasting model coupled with Chemistry) for March of four high-AOD years (2004, 2007, 2010, and 2014) and four low-AOD years (2001, 2003, 2005, 2011), as identified in Fig. 1c, in subtropical Asia (model configurations and validation are detailed in "Methods" and Supplementary Figs. 6 and 7). The difference in cloud fractions between simulations with and without BB emissions is calculated and then compared with the results from OMRs. Our model simulations successfully reproduce the aerosols' impact on cloud enhancement in southeastern Asia for both the horizontal distribution (Fig. 3a) and vertical stratification (Fig. 3c).

It is well acknowledged that aerosol, from both anthropogenic activities and BB emission, may influence cloud formation directly through absorbing/scattering solar radiation (aerosol–radiation interaction, ARI) and indirectly by serving as cloud condensation nuclei (aerosol–cloud interaction, ACI)[8,9,43–45]. To identify the crucial emission sources and key processes of aerosol in such cloud enhancement, we also tested the respective influence of anthropogenic sources (e.g., fossil fuel combustion emissions) and of ACI. The modeling results in Supplementary Fig. 8 clearly demonstrate that neither the ARI effect of fossil fuel sources nor the ACI effect of BB aerosols can substantially perturb the springtime cloud cover in subtropical Asia, confirming the dominant role of BB smoke's radiative effect on the low-cloud enhancement.

As shown in Fig. 3b, c, the prevailing westerlies transport the BB smoke plume at an altitude around 3 km, i.e., above the cloud layer, thereby producing a substantial atmospheric heating by light-absorbing aerosols, particularly BC[46–49]. The monthly averaged short-wave radiative heating can reach up to 3 K day$^{-1}$ in southern China, particularly over the land area bordering Beibu Bay (Fig. 3b), corresponding to the region of low-cloud enhancement (Figs. 1a and 3a). The substantial warming tendency above the area downwind (about 1000 km away) of the BB source region is accompanied by a substantial cooling in the PBL, particularly over the land area at the lee side of the Shan Plateau (near 105°E). Accordingly, more water vapor tends to condense onto cloud droplets under the higher relative humidity in the dimming region in the upper PBL, which are uplifted further by the converging circulations of the monsoon, and then transported eastward by the upper-level westerlies, resulting in a substantial enhancement in cloud beneath the BB smoke plume in the downwind region (Fig. 3c). It is worth noting that an increasingly thick and brighter cloud layer underneath the BB smoke plume further amplifies the PBL cooling and the heating tendency above, respectively, thereby causing a positive feedback. In contrast, in the low-AOD years with less influence of the BB emissions, such aerosol–cloud–PBL interaction and the resultant enhancement of cloud in subtropical Asia is relatively weak (Fig. 3d–f).

The vital role of aerosol–cloud–PBL interaction in cloud enhancement over the land area bordering Beibu Bay can be clearly demonstrated by the simulation and detailed analyses for a typical case on 13 March 2004. On that day, the upper-air warming and surface cooling reached up to 5 and 10 K, respectively, at Nanning, China (Fig. 4a). The simulations with ARI on/off clearly indicate a substantial low-cloud enhancement by the BB smoke, showing good agreement of the diurnal cycle of low clouds in the ARI-on case compared to both satellite and ground-based observations (Fig. 4b). A statistical analysis of the eight highest and eight lowest ARI cases (according to OMR surface air temperature) at Nanning in March 2004 suggests a substantial difference in the air temperature profile caused by the

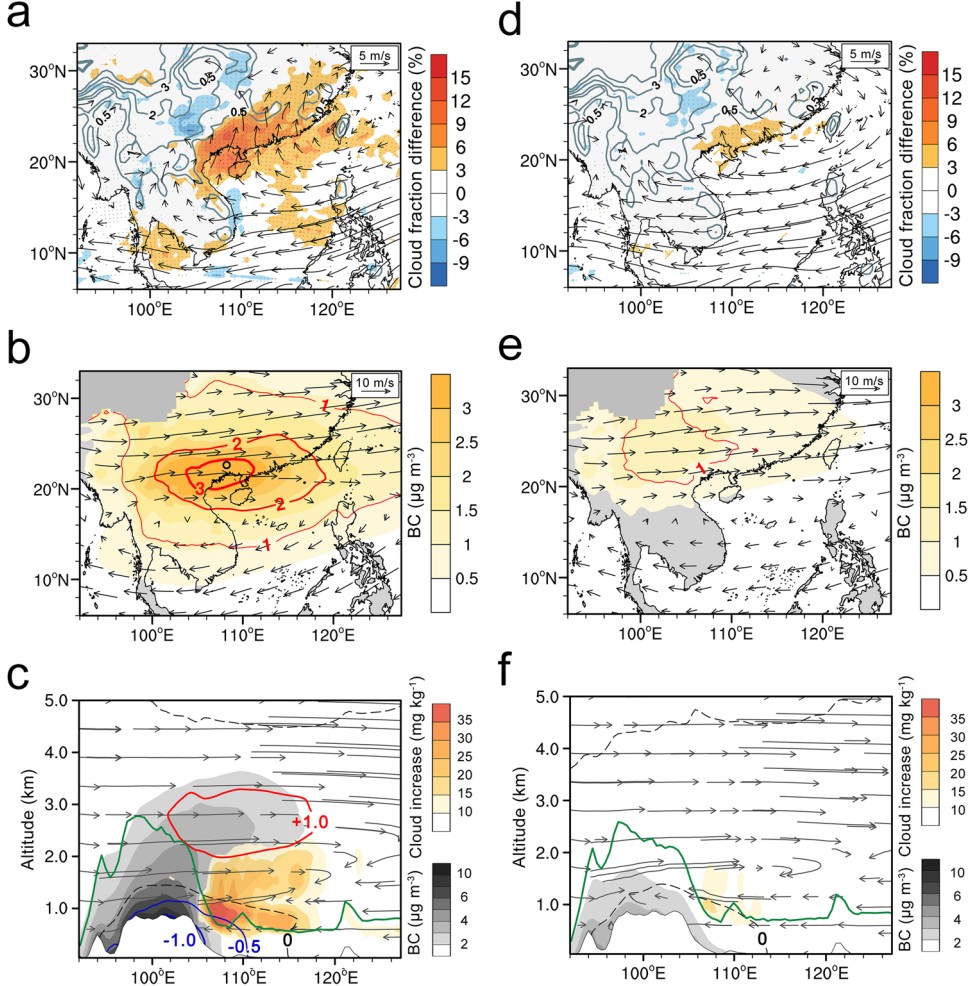

**Fig. 3 Simulated low-cloud enhancement by semi-direct effect of biomass-burning aerosols in subtropical southeastern Asia. a** Monthly averaged difference of low cloud fraction plotted with winds at an altitude of 1 km in March of high aerosol optical depth (AOD) years. **b** Spatial patterns of black carbon (BC) concentrations and aerosol-induced atmospheric heating rate at the altitude of 3 km (red isolines with the unit of K day$^{-1}$) in March of high-AOD years. **c** Vertical cross-section of BC concentration, cloud enhancement and air temperature difference (red contours for heating and blue contours for dimming, unit: K) along the coastal region (17°N–23°N) for the runs with ARI effect on/off in March of high-AOD years (2004, 2007, 2010, and 2014). **d–f**, Same as **a–c** but for low-AOD years (2001, 2003, 2005, 2011). The results are calculated from the difference between simulations with and without considering aerosol–radiation interaction (EXP_ARI and EXP_exAR). Note: Gray isolines in (**a**) and (**d**) show topography (unit: km) and black dots mark the grids passing a *T*-test. Green lines in (**c**) and (**f**) show the boundary layer height at 6:00 UTC. The black circle in (**b**) shows the location of Nanning.

aerosol–cloud–PBL interaction (Supplementary Fig. 9). Correspondingly, as illustrated in Fig. 4c, the WRF-Chem simulations show a substantial low-cloud enhancement by the smoke aloft, especially in the afternoon of the days with higher ARI. This kind of aerosol–cloud–PBL interaction is also demonstrated by a case study for 13 March 2004 at Wuzhou using a one-dimensional (1-D) WRF-Chem simulation ("Methods" and Supplementary Fig. 10).

Springtime low cloud in subtropical Asia has been demonstrated to persist below an altitude of 3 km (i.e., around 700 hPa) (Supplementary Fig. 1b) due mainly to low-level moisture convergence and wind from the South China Sea with abundant water vapor (Fig. 5a), which is strongly affected by the onset of the East Asian monsoon system[18,21,50]. The Asian winter monsoon drives an overall clockwise large-scale circulation along the coastal waters in spring, which brings a continuous supply of water vapor from the South China Sea to coastal South China and the Indochina Peninsula (Fig. 5a, c). Lagrangian dispersion modeling for air masses at different locations/altitudes of the low-cloud enhancement confirms the origin and transport pathway of water vapor (Supplementary Fig. 11). In order to further

understand the impact of BB emissions on monsoon and water vapor transport, we examined the horizontal and vertical distribution of changes in wind and water vapor between the runs with/without BB for the high-AOD years. As shown in Fig. 5b, d, under the influence of BB-induced ARI, the monsoon circulations were weakened associated with a negative anomaly of pressure in coastal South China. In addition, the westerlies between the altitudes of 1.5–3 km were also enhanced, thereby the advection of water vapor from the South China Sea to the Shan Plateau were weakened and hence more water vapor accumulated in the lower troposphere over Beibu Bay and Hainan (Fig. 5d), which could also increase the clouds. By the combined effect of the adjustment of monsoon circulation in horizontal direction and the aerosol–cloud–PBL interaction in vertical direction, the low-cloud enhancement was amplified and maintained within the middle and upper PBL in the ARI-induced dimming region over land in subtropical Asia (Figs. 3c and 5d).

The aerosol–cloud–PBL interaction coupled with the unique monsoon regime can provide an explanation for the difference compared to other regions. For example, over main BB regions like the Amazon[42], BB smoke has been proven to reduce

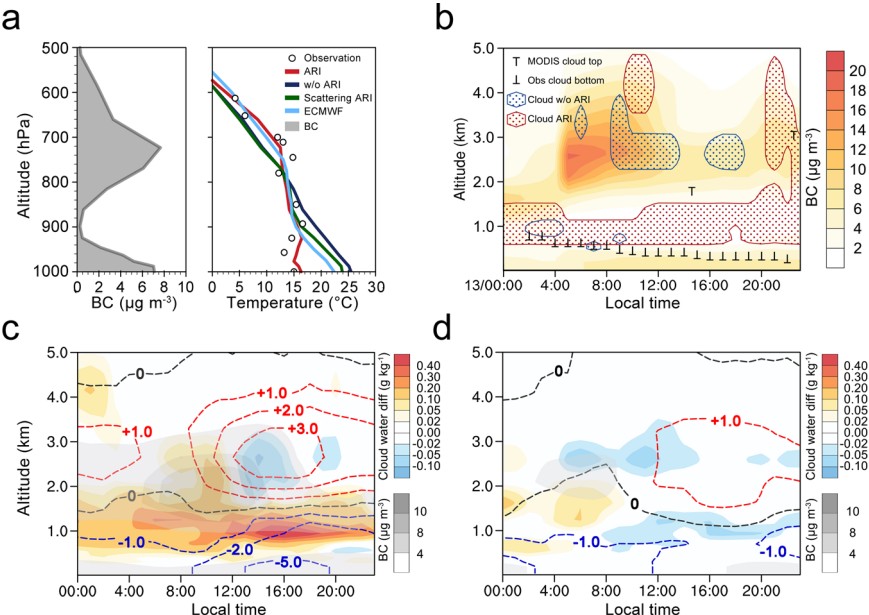

**Fig. 4 Vertical structure and diurnal evolution of black carbon (BC), air temperature, and clouds for typical episodes elucidated by model simulations. a** Vertical distribution of BC and air temperature in different modeling scenarios over Nanning (location shown in Fig. 3b) on 13 March 2004. Note that aerosol–radiation interaction (ARI), w/o ARI, scattering ARI indicate simulations with/without ARI effect, only accounting for aerosols' scattering effect, respectively, and ECMWF means European Centre for Medium-Range Weather Forecast Interim reanalysis data. **b** Diurnal cycle of cloud distributions from the runs with ARI on/off on 13 March 2004, compared with satellite and ground-based observations. Diurnal cycle of averaged cloud difference and BC concentration for the **c** 8 highest and **d** 8 lowest aerosol–cloud–boundary-layer interaction days in Nanning in March 2004. The dashed lines in (**c**, **d**) represent the air temperature difference with/without ARI effect (red for heating and blue for dimming, unit: K), and the aerosol–cloud–PBL interaction days are classified using the observation minus reanalysis (OMR) difference in surface air temperature at Nanning.

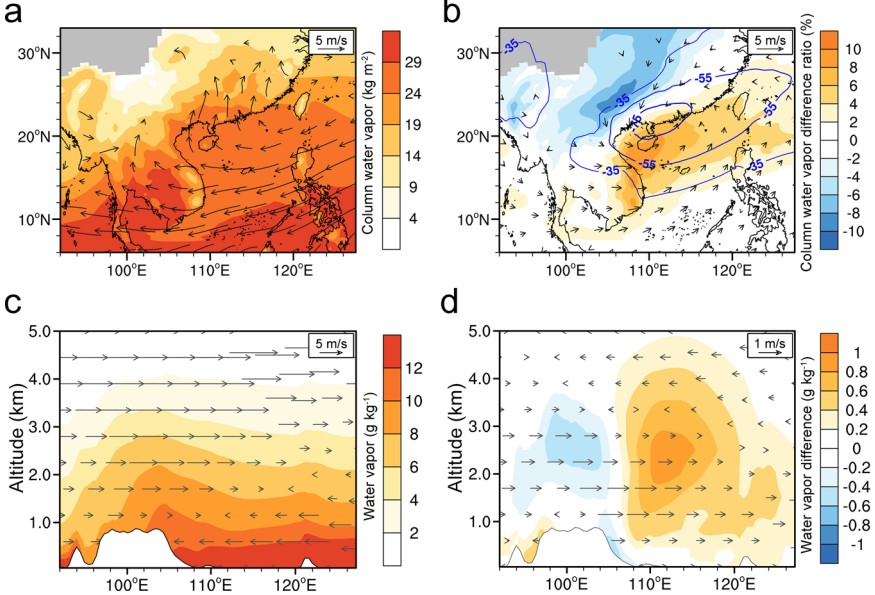

**Fig. 5 Synergetic effect of aerosol–cloud–PBL interaction coupling with the monsoon circulation in subtropical southeastern Asia elucidated by model simulations. a** Monthly averaged column water vapor below 3 km and wind field at the altitude of 1 km in March of high aerosol optical depth (AOD) years. **b** Changes of column water vapor and pressure (blue contours, Unit: Pa) and wind field at the altitude of 1 km due to biomass-burning aerosols' ARI effect. **c** Vertical cross-section of water vapor with the wind field along the coastal region (17°N–23°N) in (**a**). **d** Same as **c** but for the changes due to the radiative effect of biomass-burning aerosols. Note: Results are averaged for March of high-AOD years (2004, 2007, 2010, 2014). The radiative effect of biomass-burning aerosols is shown by differences between the experiments with and without aerosol–radiation interaction.

cloudiness via suppressed moisture fluxes due to surface cooling and a lower relative humidity driven by aerosol heating, just as predicted by the originally defined semi-direct effect of aerosol. On the other hand, the effect of BB aerosol has been shown to enhance cloudiness over the southeast Atlantic[5,14]. There, the moisture supply and PBL cooling from BB smoke affect oceanic and land areas differently (Supplementary Figs. 12 and 13) and thereby result in contrasting effects on low cloud over land and

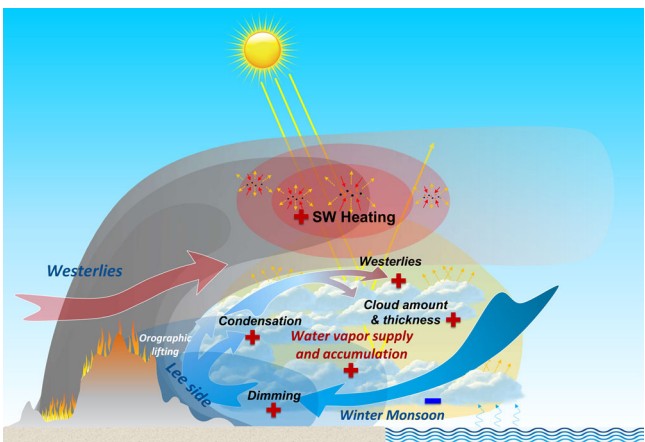

**Fig. 6 Synergetic feedback of smoke aerosol–cloud–boundary-layer interaction coupling with the monsoon in subtropical southeastern Asia.** The gray shading indicates the biomass-burning plume. The red area in the plume shows short-wave heating by absorbing aerosols like black carbon above the clouds. The blue shading on the lee side of the Shan Plateau indicates the strong dimming caused by aerosol–cloud–boundary-layer interactions over land. The yellow shading represents the water vapor supply. The plus and minus signs indicate positive and negative role in the feedback loop, respectively. SW means shortwave.

sea (Fig. 1b). However, in subtropical Asia, the semi-direct effect of BB aerosol in the vertical direction greatly enhances low cloud over ocean and land, as a result of the abundant horizontal moisture supply under the influence of adjusted monsoon circulation. This is because the air masses become saturated in the middle and upper PBL by ARI-induced cooling, causing enhanced low clouds extending from land to the oceanic area downwind. Here, the radiative heating of smoke above the clouds creates a strong inversion that is conducive to the formation of extensive shallow clouds beneath it, which are maintained by PBL cooling over land associated with the persistent transport and accumulation of water vapor by the weakened monsoon circulation (Fig. 6). Therefore, cloud cover enhancement in subtropical Asia is comparable to that over the southeast Atlantic even with much lower BB emissions, due mostly to the synergetic effect of aerosol–cloud–PBL interaction and the adjustment of monsoon circulation.

Although the adjusted monsoon circulation was also associated with the aerosol–cloud–PBL interaction at the regional scale, one might raise the question if the adjusted monsoon alone could cause this kind of cloud enhancement. To quantify the role of the monsoon change, we conducted another WRF-Chem experiment (EXP_ARIwind_exARITemp_ndg, "Methods"), in which the wind was nudged to that of the simulation with ARI effect (EXP_ARI) and the air temperature was nudged to that without ARI effect (EXP_exAR). The difference between EXP_ARIwind_exARITemp_ndg and EXP_exAR can give a quantitative estimation of the role of the adjusted monsoon circulation in the synergetic feedback. A comparison of Supplementary Fig. 14 suggests that the adjusted monsoon contributed about 25% of the enhanced low cloud, with the rest contributed mainly by the aerosol–cloud–PBL interaction.

## Discussion

Our results demonstrate that aerosol–radiation interaction caused by biomass-burning smoke dominates the springtime low-cloud enhancement in southeastern Asia. The coupling of above-cloud heating and surface dimming manifests itself in the climatology of

OMR air temperature bias for high-AOD days in spring (Supplementary Fig. 15). Given that low clouds substantially influence the radiative energy balance, the large-scale low-cloud enhancement documented here will influence the regional climate and weather conditions[19,23,44,51]. In subtropical Asia, biomass-burning smoke undergoes substantial continental-scale long-range transport[34], whereby the low-cloud enhancement covers a land area of about half a million km² with a population greater than 270 million. Given the direct impacts the clouds over land have on human activities, such as solar-energy generation, agricultural production, and regional climate, the mechanism reported in this study is important for regional sustainability and needs to be included in future forecast and assessment models.

## Methods

**OMR approach.** OMR approach is used to locate the hotspots of typical regions with cloud influenced by biomass burning. The OMR method is based on the assumption that the difference between observations and models reflects the impact of un-resolved processes, is a well-established method in atmospheric science to study anthropogenic impacts on meteorology[27,29,52–54]. Data assimilation in global reanalysis models usually tends to exclude observations with a bias above a certain threshold, e.g., 3–5 time the standard deviation of the observation errors[25,26]. It means that these real biases, which result from missing physical or chemical processes in the model, have been misinterpreted as observational errors and discarded during the data assimilation procedure for the reanalysis data[27,29]. Therefore, investigation of the difference between observation and reanalysis provides a chance to study specific processes, especially in a region with intense air pollution[27,28]. This method has been tested by previous work and proved to be well suited to identify the effects from unresolved human impacts on the lower tropospheric air temperature[27–29,52–54]. Quantitatively, in regions with intense air pollution the statistical average of OMR values might provide minimum estimations of the aerosol effects, considering that some observations with smaller departures have already been partly assimilated.

In this study, we use MODIS (Moderate-resolution Imaging Spectrometer) satellite retrievals as "observation", and ERA-Interim (European Centre for Medium-Range Weather Forecasts Interim Re-Analysis) and MERRA2 (Modern-Era Retrospective analysis for Research and Applications, Version 2) as "reanalysis". According to previous works, ERA-Interim assimilate less than 50% total radiosonde measurements[55] and MERRA2 only assimilate clear-sky satellite radiation[56], which may lead to obvious bias in comparison with observations under the condition of high aerosols loading or biased clouds[27]. Based on the OMR analysis of clouds, we could estimate the impact of BB aerosols on clouds.

**WRF-Chem simulations.** To quantitatively understand effect of aerosols on clouds and the behind mechanisms, WRF-Chem model (Version 3.6.1) is employed in this work. WRF-Chem is a chemical transport model considering online-coupled meteorological processes and chemical transformation of trace gases and aerosols. WRF-Chem can simulate the aerosol effects on weather processes and is widely used to investigate the aerosol effects on weather[57,58]. Key physical parameterization options for the model are listed in Supplementary Table 2. The simulations were conducted for the entire month of March 2001–2015 for Asia and August 2010 for Atlantic-Africa. Each run covered 48 h with the first 24 h as model spin-up and the last 24 h for the final analysis. The initial and boundary conditions of meteorological fields are updated from the 6-h NCEP (National Centers for Environmental Prediction) global final analysis (FNL) data with a 1° × 1° spatial resolution. The initial and boundary conditions of chemistry are MOZART-4 results acquired from the National Center for Atmospheric Research (NCAR). The chemical outputs from the preceding run are used as the initial conditions for the next. A similar modeling configuration and settings have been successfully adopted in our previous works and have shown good performance on reproduction of aerosol–radiation interactions[30,48,59,60].

Both natural and anthropogenic emissions are considered in this work. The BB emission is from the Quick Fire Emission Dataset (QFED), which is calculated using the FRP (top-down) approach and comprise emissions for several species[61]. The anthropogenic emission is from MIX, which includes emissions from power plants, residential combustion, industrial process, on-road mobile sources, and agricultural activities[62]. The biogenic emissions are calculated online using the Model of Emissions of Gases and Aerosols from Nature (MEGAN), including more than 20 biogenic species[63].

To investigate the impacts of BB plumes on cloud, six parallel numerical experiments were performed, with the first four and the last two aiming at the effects of aerosol radiative interaction (ARI) and aerosol–cloud interaction (ACI) on clouds, respectively: (1) a regular simulation without ARI effect (EXP_exAR), in which radiation transfer was not influenced by atmospheric aerosols, (2) a simulation with full ARI effect (EXP_ARI), in which optical properties of aerosol were calculated at each time step and then coupled with the radiative transfer model for both short- and long-wave radiation, (3) an experiment with the ARI

effect but only including anthropogenic emissions (EXP_AAR), in which BB emissions were subtracted, and (4) an experiment that only accounted for the effects of aerosol scattering by eliminating the imaginary part of aerosols (EXP_SAR), (5) a simulation including the effect of ACI from both BB and anthropogenic sources (EXP_FAC), and (6) a simulation only considering the ACI caused by anthropogenic sources (EXP_AAC). For the Atlantic-Africa region, another two parallel numerical experiments were performed to investigate the ARI of smoke on clouds: (1) a simulation without ARI effect (EXP_exAR_AA) and (2) a simulation considering BB aerosol effects on radiation transfer (EXP_ARI_AA). The anthropogenic sources in Africa were ignored because they are much smaller in comparison to BB sources. The detailed modeling setting is shown in Supplementary Tables 2 and 3.

The model simulated cloud fraction data was shown as two-dimensional one by using the COSP (Cloud Feedback Model Intercomparison Project Observation Simulator Package) approach, which facilitates the comparisons of model results with the observations in a consistent way[64]. In the COSP calculation for this study, we estimated the optical depth of clouds (COT) at different layers according to the Chang's method[65]. As the clouds are mainly distributed below an altitude of 3 km in the study region, we did not consider cloud ice in the calculation.

To demonstrate the aerosol–cloud–PBL interaction response to low-cloud enhancement, particularly the role of upper PBL clouds, we conducted additional 1-D (single column) WRF-Chem simulations at Wuzhou on 13 March 2004 (i.e., the case shown in Fig. 4a, b). The simulations were initiated with meteorological profiles over Wuzhou from the EXP_ARI simulations at 06:00 local time on 12 March (i.e., with an 18-h spin-up time). For the initial condition of the aerosol profile, the maximum profile of 13 March was used to keep the 1-D simulation similar to the overall aerosol profile on 13 March over Wuzhou. Three experiments were conducted: CEXP_exAR (without the effect of aerosols on radiation), CEXP_AR&exCR (with the effect of aerosols on radiation but without that of clouds on radiation), and CEXP_AR&CR (with the effects of both aerosols and clouds on radiation) (see Supplementary Table 4). As shown in Supplementary Fig. 10, low clouds play an important role in the interaction and were themselves enhanced in the upper PBL because of the increased relative humidity associated with the entire PBL dimming. The smoke heating above the cloud is also substantially enhanced (Supplementary Fig. 10b). These results, together with the three-dimensional (3-D) simulation presented in Figs. 3 and 4, confirm the importance of aerosol–cloud–PBL interactions in the vertical direction.

We further conducted another WRF-Chem experiment to quantify the influence of the adjusted monsoon on the synergetic feedback (EXP_ARIwind_exARITemp_ndg), in which the wind was nudged to that of the EXP_ARI simulation (i.e., the adjusted monsoon circulation) and the air temperature was nudged to that of the EXP_exAR simulation (i.e., no influence from aerosols). The difference between EXP_ARIwind_exARITemp_ndg and EXP_exAR can give a quantitative estimation of the role of the adjusted monsoon circulation in the synergetic feedback (Supplementary Fig. 14).

**Lagrangian modeling**. The transport and dispersion simulations were made using a Lagrangian dispersion model, the Hybrid Single-Particle Lagrangian Integrated Trajectory (HYSPLIT) model developed in the Air Resource Laboratory of the National Oceanic and Atmospheric Administration[66]. The model calculates the position of particles by mean wind and a turbulent transport component after they are released at the source point for forward simulation or receptor for backward run. Briefly, the model is used to conduct hourly forward or backward particle dispersion simulations. In each simulation, particles were released at the site and tracked backward in time for a 7-day period. The hourly position of each particle was calculated using a 3-D particle, i.e., horizontal and vertical method. The air concentrations were calculated according to the particle number distribution. We calculated the air concentration at a specific layer (for example 3 km altitude or 100 m altitude), which represents the distribution of the surface probability or residence time of the simulated air mass. We used the WRF-Chem simulated meteorological data to run the LPDM model, following the method described in Ding et al.[67].

**Additional data**. Satellite data. Moderate-resolution Imaging Spectrometer (MODIS) provides the daily/monthly AOD, cloud fraction, and fire location[68]. Absorbing aerosol index (AAI) is retrieved from Ozone Monitoring Instrument (OMI)[69]. Cloud-Aerosol Lidar and Infrared Pathfinder Satellite Observations (CALIPSO) data are used to show the vertical distributions of smoke extinction and cloud[70] (cloud occurrence is defined as the ratio of samples with cloud detection to all samples for each grid[71]).

Station observation data. Integrated Surface Data (ISD) archived provides the hourly surface meteorological observations, including near-surface air temperature and cloud observations. It is composed of worldwide surface weather observations from over 35,000 globally distributed stations. The Integrated Global Radiosonde Archive (IGRA) contains radiosonde and pilot balloon observations globally, which provides vertical temperature profiles. AERONET gives the absorbing aerosol optical depth (AAOD) data at 1020, 870, 675, 440 nm wavelength globally[72].

BB emission inventory. The Global Fire Emissions Database, Version 4.1 (GFED4s) provides monthly burned area and fire carbon emissions all around the globe during the time period from 1997 to 2017 (ref. [73]).

## Data availability

The reanalysis data FNL can be downloaded at website: https://rda.ucar.edu/datasets/ds083.2/. The reanalysis data ERA-Interim are available at website: https://www.ecmwf.int/en/forecasts/datasets/reanalysis-datasets/era-interim. The reanalysis MERRA2 data can be obtained at website: https://disc.gsfc.nasa.gov/datasets?project=MERRA-2. The satellite data like MODIS, OMI, and CALIPSO data are available at website: https://search.earthdata.nasa.gov/. AERONET data can be obtained at website: https://aeronet.gsfc.nasa.gov/. ISD data are available at website: https://www.ncei.noaa.gov/products/land-based-station/integrated-surface-database. Sounding data can be downloaded from website: ftp://ftp.ncdc.noaa.gov/pub/data/igra. The biomass burning emission data QFED are available at website: http://ftp.as.harvard.edu/gcgrid/data/ExtData/HEMCO/QFED, and the biomass burning emission data GFED can be downloaded from website: https://www.geo.vu.nl/~gwerf/GFED/GFED4/. The anthropogenic emission data MIX are available at website: http://meicmodel.org. Additional data related to the modeling results are available at figshare data publisher: https://doi.org/10.6084/m9.figshare.15073974.

## Code availability

The source code of the WRF-Chem model is archived on UCAR data repository (http://www2.mmm.ucar.edu/wrf/users/download). The HYSPLIT model can be acquired from the NOAA Air Resources Laboratory (http://www.ready.noaa.gov). Data processing techniques are available on request from the corresponding authors.

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

## Acknowledgements

The work is supported by National Natural Science Foundation of China (41725020 and 41922038) and the CSC-China Scholarship Council for joint Ph.D. grant awarded to K.D. This study was partly supported by European Research Council via ATM-GTP 266 (742206), Academy of Finland Center of Excellence in Atmospheric Sciences (grant number: 272041), Academy of Finland Flagship "Atmosphere and Climate Competence Center (ACCC: grant number 337549), and Fundamental Research Funds for the Central Universities (grant number: DLTD2107). We are grateful to the High-Performance Computing & Massive Data Center (HPC&MDC) of School of Atmospheric Science, Nanjing University for doing the numerical calculations in this paper on its Blade cluster system. We thank Prof. Lili Lei at Nanjing University for insightful discussions on the data assimilation of ERA-Interim and Zhoukun Liu for his help on COSP. We acknowledge the use of imagery from the NASA Worldview application, part of the NASA Earth Observing System Data and Information System (EOSDIS).

## Author contributions

A.D., Y.C., and M.O.A. conceived the overall idea, K.D. and X.H. performed most of the analysis and model simulations, M.W., Z.W., and D.Z. analyzed satellite data, A.D., Y.C., K.D., X.H., and M.O.A. drafted the paper. H.S., V.-M.K., T.P., Z.T., J.S., H.L., H.W., R.W., P.Z., D.R., M.K., U.P., and C.F. edited the paper. All authors discussed and revised the paper.

## Competing interests

The authors declare no competing interests.
