## [Peer Review file · Nature Communications]

REVIEWER COMMENTS

Reviewer #1 (Remarks to the Author):

Review of: Asian monsoon amplifies semi-direct effect of biomass burning aerosols on low cloud formation

This paper uses comparison of satellite observations with a reanalysis together with a series of numerical simulations to examine the effect of biomass burning (BB) aerosols on boundary-layer clouds over Asia. It demonstrates that BB aerosols derive a significant increase in cloudiness over this region. The cloud cover increase is as large as in the southeast Atlantic despite a weaker aerosol perturbation. This large cloud sensitivity to BB aerosol is argued to emerge due to the monsoon circulation, which brings large amounts of water vapor from the ocean into the dimming region over land. The moist air becomes colder by the aerosol dimming effect and hence saturated, while the shortwave absorption above the cloudy-layer generates stronger inversion, which further amplifies the cloud cover.

The topic is scientifically and socially interesting. However, I have two major comments and two minor suggestions that need to be addressed before the paper will be ready, in my opinion, for publication.

1) A major part I missed in this paper concerns the effect of aerosol forcing on the monsoon circulation. It is known that aerosol forcing can affect the monsoon circulation¹⁻³, which may feed-back to the aerosol effect on clouds. However, this is completely ignored here. The WRF simulations should be able to identify the changes in the circulation. Specifically of interest here is whether the aerosol forcing change the water vapor supply inland?

It is mentioned that the results are due to "synergetic effect of aerosol-cloud-PBL feedbacks and the large-scale monsoon circulation", however, the full possible synergy is not explored and only small part of it is presented here. Some exploration of the aerosol-cloud-circulation coupling, in a paper that mentions all of the components in its title, is required.

2) I think that the approach of comparing satellite observations with reanalysis is flawed, specifically in this case. Taking the difference between observations and reanalysis (which assimilate part but not all of the observations) is not a reliable way to identify aerosol effects. This is particularly true for the semi-direct effect, where simple radiosonde measurements (which are assimilated) effectively introduce the relevant aerosol effect into the reanalysis. I appreciate that this method was used before, but I can't really see any reason to trust it in this case. Some further justifications are needed. For example, what is the fraction of observations (and specifically radiosonde measurements) that were excluded due to "bias" over this region? Were the conditions measured in these excluded measurements consistent with the semi-direct effect? etc.

If the editor chooses to move forward towards a possible publication, I recommend that some effort is made to justify the OMR approach, specifically for the case of aerosol semi-direct effect, and (even more important in my opinion) to explore the possible role of circulations changes.

Besides that, I have two more minor comments:

1. There are too many references to different locations with data or figures and the order of the figures could be organized differently so that it will be easier to follow. An example from L 112: "(Extended Data Fig. 4 and 7 and Supplementary Fig. 5)". The reader is referred to 3 different figures at two different locations to make a single point. It is a bit exhausting and hard to follow. I am sure it could be organized differently (even with the limitation of figures in the main text).

2. L204-208: too long and complicated sentence.

References

- 1 Li, X., Ting, M., Li, C. & Henderson, N. Mechanisms of Asian summer monsoon changes in response to anthropogenic forcing in CMIP5 models. *Journal of Climate* 28, 4107-4125 (2015).
- 2 Li, Z. et al. Aerosol and monsoon climate interactions over Asia. *Reviews of Geophysics* 54, 866-

929 (2016).

3 Lau, K. M. & Kim, K. M. Observational relationships between aerosol and Asian monsoon rainfall, and circulation. *Geophysical Research Letters* 33, doi:10.1029/2006gl027546 (2006).

Reviewer #2 (Remarks to the Author):

This paper presents an intriguing hypothesis that the monsoon circulation in Asia contributes to a strong increase in cloud cover over subtropical Asia resulting from semi-direct forcing of biomass burning aerosols. The authors present empirical results based mainly on the differences between satellite observations and a model reanalysis that lacks aerosol radiative effects, as well as simulations with an atmospheric model applied with and without aerosols. While the empirical results are certainly suggestive of a meaningful effect, I found the discussion of the actual physical processes to be limited and the evidence supporting the physics to be weak and/or not clearly demonstrated. Part of the difficulty in clearly supporting the physical arguments is the structure of the paper, which includes extensive analysis and discussion of semi-direct aerosol effects in other regions, most of which have already been documented in the literature, and also the use of both extended data and supplementary information, which leads the reader to shuffle around in two different documents to follow the argument. In its present form, the paper is not convincing enough for publication. It might be suitable for publication after major revision that offers both a clearer presentation of the evidence supporting the hypothesized physics associated with the Asian monsoon and an adjustment of the structure of the paper that focuses on that specific argument.

In the discussion of the application of their empirical techniques to the southern Africa/southeast Atlantic Ocean case (lines 118-125) the authors argue that the strong contrast between the ocean and terrestrial response of low clouds to biomass burning aerosols are attributable to the "aerosol absorption cloud fraction feedback" and the "semi-direct effect of absorbing aerosols", respectively. However, it is important to note that the original concept of the "semi-direct" effect was a reduction in cloud cover due to radiative heating of the air (Hansen et al. 1997). In that sense, it is a bit confusing to argue that the response over land is something other than a "semi-direct" effect. Do the authors intend to suggest that the "aerosol absorption cloud fraction feedback" is something different from a "semi-direct effect"? Furthermore, calling a process a feedback is typically reserved for a process where the response of the system to a forcing further affects the magnitude of the forcing. In what way is the proposed process over land a feedback? The reference provided for this "aerosol absorption cloud fraction feedback" is to a paper based on aerosol-cloud interactions over the Amazon, otherwise there is a long list of references purported to support the notion of a land/ocean contrast in the cloud response over southern Africa and the adjacent Atlantic Ocean. Is there a paper in the literature that provides evidence for the specific physical process leading to the reduction in cloud cover over land in Africa?

The argument for why the semi-direct effect may be stronger over subtropical Asia compared to the southeast Atlantic Ocean is attributed by the authors to stronger transport of low-level moisture into the cloud layer due to the monsoon circulation, which seems plausible. However, there is little discussion of the actual physics of the semi-direct effect in this region; or what the authors have labeled (but not really fully described) the "smoke-cloud-PBL feedback coupling". The increase in cloud over the southeast Atlantic Ocean has been attributed in the past to a suppression of turbulent mixing of humidity from the cloudy boundary layer to the free troposphere above the cloud layer (Johnson et al 2004, Wilcox 2010). Are the authors arguing that the same process is occurring over subtropical Asia? If so, I think that the authors will have to consider some of the details of the vertical structure in the region: in particular the relation of the cloud top height to the PBL height. Over the southeast Atlantic it is often the case (though not always) that the smoke is entirely, or at least predominantly, located in the free-troposphere and the clouds are capped by the strong subtropical inversion capping the marine boundary layer. This vertical structure is important for the proposed mechanism in that location because of the effect of

stabilization of the lower free-troposphere inhibiting ventilation of the boundary layer through turbulent entrainment. None of the relevant physics of these processes or others that the authors might be arguing for over subtropical Asia is discussed.

The cross-sections through the simulated aerosol and cloud distributions in figures 3b and 3d are helpful for conveying the relation between the aerosols and clouds, but also raise many questions. First, the authors refer in line 147 to the "biomass burning source region", but none of the details of the geography of the sources is discussed. Where exactly is the source region? What are the optical properties of the aerosols and are there published estimates of the magnitude of the radiative heating rate? Is the magnitude of the atmospheric temperature difference shown in figure 3b consistent with existing estimates of the shortwave radiative heating rate? All of these details are essential to arguing that the aerosols might be motivating a semi-direct effect and that the model simulations are representative of the realistic aerosol radiative heating in the atmosphere, yet none of this is discussed.

Second, the smoke over the source region (shown in fig. 3b) is at the same elevation as what I presume is the base of the cloud layer, or at least the lowest elevation where an enhancement of cloud is indicated to the east of the source region. For the semi-direct effect to be operating the smoke must be elevated above the cloud layer. The authors argue that monsoon convergence lofts the smoke above the cloud layer, but the only evidence presented to support this are small wind vectors in figures 3a and 3c. Perhaps there is literature that documents the link between the dynamics and the vertical profile of the smoke? Presumably mixing of the smoke with the cloud layer would also result in microphysical indirect effects that might be commingled with the semi-direct effects. Indeed, the cross section in extended data fig. 6a seems to imply that the smoke and clouds may be mixed at least as far east as 110 deg. E. Thus, I was struggling to reconcile the extinction cross-section in extended data fig. 6a suggesting strong coincidence of the smoke and cloud with the extinction profiles in fig. 2 which are presented to argue that the smoke is predominantly above the cloud. It is also unclear how then the authors can be sure that some of the effects they are seeing are not attributable to microphysical interactions of aerosols with clouds?

The crux of the authors' argument is that the monsoon circulation acts to amplify the semi-direct enhancement of cloud, as stated in the title. I found it rather difficult to follow the evidence for this. First, it is argued that the circulation "brings large amounts of water vapor ... [to] the lee side of the plateau"(line 178). The referenced figure in the extended data shows the circulation, but the transport of moisture is not presented quantitatively, so while this seems highly plausible, it is not demonstrated. I also note that the "plateau" is referenced a half-dozen times throughout the paper, but it is never made clear exactly which plateau is being referred to.

Next it is argued that the "air masses become saturated in the middle and upper PBL by ARI-induced cooling" (line 180). But the figures referenced for this indicate cooling only near the surface and apparently below the level of enhanced cloudiness shown in figure 3b, so it was not clear to me from the evidence provided that the cooling is truly where the clouds are being enhanced. Also, since there is no discussion of the PBL structure, or its relation to the aerosol and clouds (as noted above) it is hard to know exactly what the authors mean when they say "middle and upper PBL".

Next it is argued that "radiative heating of smoke above clouds creates a strong inversion that is conducive to the formation of extensive shallow clouds below it" (line 183). But the only figure referenced for this is a cartoon graphic (fig. 4). There are some temperature profiles presented in the extended figures, but there is no discussion of any details regarding how the aerosol heating is impacting the vertical temperature structure, or where the inversion is expected to be relative to the aerosols and clouds in this region, or the magnitude of any strengthening of the inversion. Furthermore, there is no discussion of the relevant physics by which one might expect an enhanced inversion to promote shallow cloud development.

Finally, there is a single mention of the shallow clouds being "maintained by the radiative cooling over land associated with the transport of water vapor by the monsoon circulation" (line 185), again referencing the cartoon rather than a paper (of which there are presumably many in the

monsoon literature) or a quantitative argument based on the authors' analysis or modeling. This link to the circulation is what I presume the authors mean when they say that the Asian monsoon "amplifies" semi-direct effects in the paper's title. In the discussion it is not presented as an "amplification", rather it is described as a "synergetic effect of aerosol-cloud-PBL feedbacks and the large-scale monsoon" (line 190). But presumably they mean that the aerosol-cloud-PBL dynamics feeds back ON the monsoon circulation? It is not really clear which elements of this system are argued to be acting as feedbacks, or exactly how these elements are acting synergistically.

I get a sense that there is the nugget of an interesting idea in this paper, but the evidence is simply not presented in a quantitatively convincing manner. Furthermore, the language throughout the manuscript, from the geography, to the physics, to buzzwords like "synergy" and "feedback", are simply not used precisely enough to make clear what exactly is being argued to be occurring. Part of the problem may be that the overall structure of the paper is ill suited to a letter journal owing to the fact that the authors have tried to present results from multiple regions with differing responses and then build their argument for why the subtropical Asian monsoon region is different. Although the structure of the argument is compelling, the authors have included a large number of figures to compare and contrast the detailed differences in the arrangement of aerosols and clouds in the observations in multiple locations and the model simulations of the monsoon system. The many references to both extended data and supplementary content begs the questions of why a letter journal was considered the right choice for a paper requiring so many figures but also subjected to such a limiting maximum word count. In many cases there are declarative statements with references to multiple figures, but little description of what the figure depicts or explanation of the relevant physics at work. One strategy to improve upon this could be to simply use the existing literature to back up claims about semi-direct effects in regions outside of subtropical Asia and then focus the discussion on describing in better detail the authors' arguments for the nature of semi-direct effects in subtropical Asia and their relationship to the monsoon circulation. Choosing a journal that allows more words and figures would perhaps make that easier to accomplish

Additional recommendation:

The authors claim on line 88 and 89 that the emissions of biomass burning in Asia are 30% of that in southern Africa; a point that was also mentioned in the abstract. The implication seems to be that the semi-direct effect is somehow more efficient, per unit of emissions in Asia, although the authors do not explicitly make that point. This result is based on emissions estimates in units of mass of carbon per unit area. From a land-use perspective, perhaps it might be meaningful to relate emissions to an aerosol-cloud effect in this way, however, the semi-direct effect responds to the radiative forcing of the aerosol, not its emissions rate. There are many other factors that could lead to different radiative forcing for the same rate of emissions, including the radiative properties of the particles, their residence time in the atmosphere, the concentration of particles in the resulting atmospheric plume, etc... This point is only incidental to the main goal of the paper, however, it is included in the abstract of the paper. If the authors are going to emphasize this point, then they should clarify exactly what the reader should be taking away from this result and why emissions rate is the right quantity to compare between the two regions. If indeed the point is to suggest that the cloud enhancement is more effective over subtropical Asia, then I think they need to make a more quantitative case to back that up.

References:

Hansen, J., Sato, M., and Ruedy, R.: Radiative forcing and climate response, *J. Geophys. Res.*, 102, 6831–6864, 1997.

Johnson, B. T., Shine, K. P., and Forster, P. M.: The semi-direct aerosol effect: impact of absorbing aerosols on marine stratocumulus, *Q. J. Roy. Meteorol. Soc.*, 130, 1407–1422, 2004.

Wilcox, E. M.: Stratocumulus cloud thickening beneath layers of absorbing smoke aerosol. *Atmos. Chem. Phys.*, 10, 11769–11777, doi:10.5194/acp-10-11769-2010, 2010.

Response to Referee #1

This paper uses comparison of satellite observations with a reanalysis together with a series of numerical simulations to examine the effect of biomass burning (BB) aerosols on boundary-layer clouds over Asia. It demonstrates that BB aerosols derive a significant increase in cloudiness over this region. The cloud cover increase is as large as in the southeast Atlantic despite a weaker aerosol perturbation. This large cloud sensitivity to BB aerosol is argued to emerge due to the monsoon circulation, which brings large amounts of water vapor from the ocean into the dimming region over land. The moist air becomes colder by the aerosol dimming effect and hence saturated, while the shortwave absorption above the cloudy-layer generates stronger inversion, which further amplifies the cloud cover.

The topic is scientifically and socially interesting. However, I have two major comments and two minor suggestions that need to be addressed before the paper will be ready, in my opinion, for publication.

Response: We appreciate the overall encouraging comments and suggestions. We revised the manuscript according to the comments and listed point-by-point responses below.

1) *A major part I missed in this paper concerns the effect of aerosol forcing on the monsoon circulation. It is known that aerosol forcing can affect the monsoon circulation¹⁻³, which may feed-back to the aerosol effect on clouds. However, this is completely ignored here. The WRF simulations should be able to identify the changes in the circulation. Specifically of interest here is whether the aerosol forcing change the water vapor supply inland?*

It is mentioned that the results are due to “synergetic effect of aerosol-cloud-PBL feedbacks and the large-scale monsoon circulation”, however, the full possible synergy is not explored and only small part of it is presented here. Some exploration of the aerosol-cloud-circulation coupling, in a paper that mentions all of the components in its title, is required.

Response: We greatly appreciate this suggestion that motivated us to further explore the impact of aerosol forcing to the monsoon circulation. Indeed, we found that this process did weaken the overall monsoon circulations and further enhanced the water

vapor supply, promoting the accumulation of water vapor in the lower troposphere of the low-cloud enhancement areas. The new results complete the whole synergetic effect of aerosol-cloud-PBL-monsoon feedback and make our discussions more comprehensive.

In the revised manuscript, we added the following figure (Fig. R1) as Fig. 5 and a paragraph to discuss these results. Also, we modified the conceptual scheme (Fig. 4 in original submission) to include the role of monsoon feedback (Fig. R2, Fig. 6 in the revised manuscript).

Fig. R1 | Synergetic effect of smoke-cloud-PBL feedback coupling with the monsoon circulation in subtropical southeastern Asia elucidated by WRF-Chem simulations. **a**, Monthly averaged column water vapor below 3 km and wind field at the altitude of 1 km in March at high AOD. **b**, Changes of column water vapor and pressure (blue contours, Unit: Pa) and wind field at the altitude of 1 km due to BB aerosols ARI effect. **c**, Vertical cross-section of water vapor with the wind field along the coastal region (17°N-23°N) in (a). **d**, Same as (c) but for the changes due to BB aerosols ARI effect. Note: Results are averaged for March of high AOD years (2004, 2007, 2010, 2014). The BB aerosols ARI effect are shown by differences between the experiments EXP_FAR and EXP_exAR.

In the revised manuscript, we added the following two paragraphs to discuss the new results:

Springtime low cloud in subtropical Asia has been demonstrated to persist under the altitude of 3 km (i.e., around 700 hPa) (Fig.1a) due mainly to low-level moisture convergence and wind from the South China Sea with abundant water vapor (Fig. 5a), which is strongly affected by the onset of East Asian monsoon systems. The Asian winter monsoon drives an overall clockwise large-scale circulation along the coastal waters in spring, which brings a continuous supply of water vapor from the South China Sea to the coastal South China and Indochina Peninsula (Fig. 5a, c). Lagrangian dispersion modelling for air masses at different locations/altitudes of the low-cloud enhancement all confirm the origin and transport pathway of water vapour (Supplementary Fig. 10). In order to further understand the impact of BB emissions on monsoon and water vapour transport, we examined the horizontal and vertical distribution of changes in wind and water vapor between the runs with/without biomass burning for the high AOD years. As shown in Fig. 5b and 5d, under the influence of BB-induced ARI, the monsoon circulations were weakened associated with a negative anomaly of pressure in the coastal South China. In addition, the westerlies between the altitudes of 1.5-3 km were also enhanced, thereby the advection of water vapor from South China Sea to the Shan Plateau were weakened and hence more water vapour accumulated in the lower troposphere over Beibu Bay and Hainan (Fig. 5d). With a combined effect of horizontal transport by monsoon and the vertical aerosol-cloud-PBL feedbacks, the low-cloud enhancement was amplified and maintained within the middle and upper PBL in the ARI-induced dimming region over land in subtropical Asia (Fig. 6).

The aerosol-cloud-PBL feedbacks coupled with the unique monsoon regions could provide an interpretation on the difference compared to other regions. For example, over main biomass burning regions like Amazon40, BB smoke has been proven to reduce cloudiness via suppressed moisture fluxes due to surface cooling and a lower relative humidity driven by aerosol heating, just as the originally termed semi-direct effect of aerosol. Similarly, the semi-direct effect of BB aerosol has been also postulated to thicken low cloud in southeast Atlantic. The moisture supply and PBL cooling take effect in oceanic and land areas separately (Supplementary Figs. 11 and 12), which thereby result in contrast effect on low cloud over the land and sea (Fig.1b). However, in subtropical Asia, the semi-direct effect of BB aerosol greatly enhances low cloud as an abundant moisture supply under the influence of monsoon. This is because the air masses become saturated in the middle and upper PBL by ARI-induced cooling, causing enhanced low cloud cover extending from land to the oceanic area downwind. Here, the radiative heating of smoke above the clouds creates a strong inversion that is conducive to the formation of extensive shallow clouds beneath it, which are maintained by PBL cooling over land associated with the persistent transport and accumulation of

water vapor by the weakened monsoon circulation (Fig. 6). Therefore, cloud cover enhancement in subtropical Asia is comparable to that in southeast Atlantic even with much less BB emission, due mostly to a synergetic effect of aerosol-cloud-PBL-monsoon feedbacks.

Fig. R2 | Mechanism of smoke-cloud-PBL feedback coupling with the monsoon in subtropical southeastern Asia. The gray shading indicates the biomass-burning plume. The red color in the plume shows shortwave heating by absorbing aerosols like BC above the clouds. Blue shading on the lee side of the Shan Plateau indicates the strong dimming caused by aerosol-cloud-PBL interactions over land. Yellow shading represents the water vapor supply. The “+” and “-” symbols indicate positive and negative feedback, respectively.

2) *I think that the approach of comparing satellite observations with reanalysis is flawed, specifically in this case. Taking the difference between observations and reanalysis (which assimilate part but not all of the observations) is not a reliable way to identify aerosol effects. This is particularly true for the semi-direct effect, where simple radiosonde measurements (which are assimilated) effectively introduce the relevant aerosol effect into the reanalysis. I appreciate that this method was used before, but I can't really see any reason to trust it in this case. Some further justifications are needed. For example, what is the fraction of observations (and specifically radiosonde measurements) that were excluded due to "bias" over this region? Were the conditions*

measured in these excluded measurements consistent with the semi-direct effect? etc. If the editor chooses to move forward towards a possible publication, I recommend that some effort is made to justify the OMR approach, specifically for the case of aerosol semi-direct effect, and (even more important in my opinion) to explore the possible role of circulations changes.

Response:

We appreciate the reviewer's comment on the OMR analysis in this work. We used the OMR approach to introduce the overall story and identify the hotspots, but not as the main method to quantify the impact.

First, for air temperature, as mentioned, the OMR approach is a well-established method in studying anthropogenic impacts (e.g., aerosol and urbanization) on meteorology (Kalnay et al., 2003; Zhao et al., 2014; Huang et al., 2018). Plenty of papers have been published already to demonstrate the impact of atmospheric aerosols, including several published by our research group with evidence based on long-term measurements (e.g., Huang et al., 2018). We understand that the OMR might not be an appropriate method for cloud study. However, for the semi-direct effect, which is mainly driven by thermal processes, the OMR analysis potentially could be used. We emphasize, however, that we used the OMR method just to qualitatively locate the "hot spots", in comparison with Atlantic-Africa, for further mechanism diagnoses and qualification of the semi-direct effect. The proposed new mechanism "Asian monsoon amplifies semi-direct effect of biomass burning aerosols on low cloud formation" stands without the OMR method. It is proven by the vertical measurements from CALIPSO Lidar (from both climatology and year-to-year difference) and WRF-Chem modelling with comprehensive supporting observational data. On the other hand, our analysis from interannual variability, CALIPSO vertical data, and WRF-Chem modeling analysis provides further evidence and a mechanistic explanation on why the OMR methods could be a promising method for studies of aerosols' impact on the semi-direct effect, which is strongly associated with vertical air temperature profiles (for the latter the OMR method has been demonstrated to stand well).

All observations used in ERA-Interim are subject to quality control and data selection, including gross error checks, systematic error checks, satellite channel blacklisting, data thinning and first guess check. Observations that contributed to the final analyses are processed by the variational quality control (VarQC). Most near-surface observations and most satellite radiances that are over land, high terrain, and

contaminated by rain are first excluded. Then the first guess check evaluates the observation departure, which is the distance between an observation and the model prior transformed at the location and time of the observation. Observations with departures more than 5-6 standard deviations of the observation errors are rejected. Observations that pass the first guess check but with large departures are given lower weights in the analyses through the VarQC with a Huber norm. Therefore, observations whose departures from the model priors exceed the prescribed thresholds are not assimilated or have only slight influences on the analyses. So, the OMR method is valid to shed light on the aerosol effects, especially for high levels of air pollutions.

We understand that the reviewer's point is that it will be better to identify the parts of sounding observations that were included and excluded for this region. Unfortunately, this can only be done by the ERA-Interim team, if all the quality control check files were recorded. However, according to our comparison with available soundings in this region (Supplementary Fig. 13) and our other previous studies in polluted regions (Ding et al., 2013; Huang et al., 2018), we found that the ARI effect by the aerosol might be strong enough to exclude most of the sounding measurements in the lower troposphere, especially for episode days with higher AOD.

Anyway, we appreciate the point that was raised by the reviewer. Taking the advice, we have added more discussions to justify the OMR results in both main text and methods. Please refer to Lines 71-76 and 322-325 in the revised manuscript, which are listed below.

L71-76: Since all observations used in the ERA-Interim are subject to quality control and data selection, observations whose departures from the model priors exceed the prescribed thresholds (e.g., several time standard deviation of the observation errors) are not assimilated or have no influences on the analyses. Therefore, the OMR approach can shed light on the aerosol effects that haven't been included in the reanalysis.

L322-325: Quantitatively, in regions with intense air pollution a statistical average of OMR value might provide minimum estimations of the aerosols' effect considering that some observation with less departures have already been partly assimilated.

References:

Dee, D. P. *et al.* The ERA-Interim reanalysis: configuration and performance of the data

assimilation system. *Q. J. R. Meteorol. Soc.* **137**, 553-597 (2011).

Ding, A. J. *et al.* Intense atmospheric pollution modifies weather: a case of mixed biomass burning with fossil fuel combustion pollution in eastern China. *Atmos. Chem. Phys.* **13**, 10545-10554 (2013).

Huang, X., Wang, Z. L., Ding, A. J. Impact of Aerosol-PBL interaction on haze pollution: multiyear observational evidences in North China. *Geophys. Res. Lett.*, **45**, 8596-8603 (2018)

Kalnay et al., Impact of urbanization and land-use change on climate, *Nature* **423**, 528-531, 2003.

Tavolato, C. & Isaksen, L. ERA report series: Data usage and quality control for ERA-40, ERA-Interim and the operational ECMWF data assimilation system. *ERA Report Series* **44** (2010).

Zhao et al., Strong contributions of local background climate to urban heat islands, *Nature* **511**, 216-219, 2014.

Besides that, I have two more minor comments:

1. There are too many references to different locations with data or figures and the order of the figures could be organized differently so that it will be easier to follow. An example from L 112: “(Extended Data Fig. 4 and 7 and Supplementary Fig. 5)”. The reader is referred to 3 different figures at two different locations to make a single point. It is a bit exhausting and hard to follow. I am sure it could be organized differently (even with the limitation of figures in the main text).

Response: We highly appreciate the reviewer’s suggestion. In the revision, the figures and text have already been re-organized through streamlining and integration. The specific modifications include:

1. Two figures, Fig. 4 and Fig.5, are added in the revised manuscript. These two figures clearly demonstrate the monsoon circulation, moisture supply, and aerosols’ radiative effect on temperature and cloud stratification.

2. According to the figure guidelines of Nature Communications, the Extended Data Figures and Supplementary Figures in the first-round submission were integrated and simplified into one Supplementary Information file. All the supplementary figures have been refined to avoid multiple figure citations for one single point.

2. L204-208: *too long and complicated sentence.*

Response: Thanks for your comment. We have rephrased this statement to “In subtropical Asia, biomass burning smoke could experience substantial continental-scale long-range transport, whereby the low-cloud enhancement covers a land area of about half a million km² with a population greater than 270 million”. Please see Lines 299-302.

Response to Referee #2

This paper presents an intriguing hypothesis that the monsoon circulation in Asia contributes to a strong increase in cloud cover over subtropical Asia resulting from semi-direct forcing of biomass burning aerosols. The authors present empirical results based mainly on the differences between satellite observations and a model reanalysis that lacks aerosol radiative effects, as well as simulations with an atmospheric model applied with and without aerosols. While the empirical results are certainly suggestive of a meaningful effect, I found the discussion of the actual physical processes to be limited and the evidence supporting the physics to be weak and/or not clearly demonstrated. Part of the difficulty in clearly supporting the physical arguments is the structure of the paper, which includes extensive analysis and discussion of semi-direct aerosol effects in other regions, most of which have already been documented in the literature, and also the use of both extended data and supplementary information, which leads the reader to shuffle around in two different documents to follow the argument. In its present form, the paper is not convincing enough for publication. It might be suitable for publication after major revision that offers both a clearer presentation of the evidence supporting the hypothesized physics associated with the Asian monsoon and an adjustment of the structure of the paper that focuses on that specific argument.

Response: We would like to thank the referee for the very helpful comments and suggestions, which do help us improve the present work. In the revised version, we have made a major revision by removing unnecessary figures and discussions and presenting a clearer story of the synergetic interaction between aerosol-cloud-PBL and monsoon circulation.

In the discussion of the application of their empirical techniques to the southern Africa/southeast Atlantic Ocean case (lines 118-125) the authors argue that the strong contrast between the ocean and terrestrial response of low clouds to biomass burning aerosols are attributable to the “aerosol absorption cloud fraction feedback” and the “semi-direct effect of absorbing aerosols”, respectively. However, it is important to note that the original concept of the “semi-direct” effect was a reduction in cloud cover due to radiative heating of the air (Hansen et al. 1997). In that sense, it is a bit confusing to argue that the response over land is something other than a “semi-direct” effect. Do the authors intend to suggest that the “aerosol absorption cloud fraction feedback” is something different from a “semi-direct effect”?

Response: Thanks for pointing out this issue. The “semi-direct” effect was first proposed by Hansen et al. (1997), which originally referred to the phenomenon that light-absorbing aerosol in the lower layers of the atmosphere causes a positive climate forcing by reducing cloud. In the context of the article (Hansen et al., 1997), the “semi-direct” effect was specially defined to distinguish itself from the indirect effect of aerosol via altering cloud microphysics (Page 6858). The most extensively acknowledged cloud response to the radiative heating by aerosol prior to 1997 is that of evaporation of clouds due to the warming tendency of the cloud layer. Thereafter, the meaning of “semi-direct” effect of aerosol has been greatly extended with varying cloud responses via different physical processes, including reduction/enhancement of low to mid-level cloud cover with a positive/negative climate radiative effect (Koch & Del Genio, 2010). The reviewer paper by Koch & Del Genio (2010) also discussed the definition: “Atmospheric BC suspended near clouds has been thought to contribute to cloud evaporation, originally termed the “semi-direct effect” (Hansen et al., 1997). This loss of cloud cover exacerbates the warming impact of BC. However, there are numerous studies that describe additional mechanisms whereby BC may either reduce or increase cloud cover, and thus there may be multiple semi-direct effects.”

Ramanathan et al. (2001) also indicated that aerosol absorption decreases the absorbed solar radiation at the surface and increases low-level static stability, leading to lower surface moisture fluxes and a smaller probability of cloud formation. However, when the absorbing aerosol resides above a low cloud deck, the absorption of sunlight by the aerosols causes a reduction in cloud-top entrainment, leading to a thickening of the cloud deck and also a negative semi-direct forcing (Johnson et al., 2004; Wilcox, 2010; Lu et al., 2018). Thus, in our manuscript we use the broader definition of the “semi-direct” effect, which means that the aerosols influence clouds by directly modifying the radiative heating rate and thermodynamic structure (Johnson et al., 2004; Koch & Del Genio, 2010).

Therefore, while discussing the southern Africa/southeast Atlantic Ocean case in this manuscript, both the cloud enhancement over ocean attributed to weakening cloud-top entrainment of overlying dry air and the reduction of cloudiness over land by the aerosol absorption cloud fraction feedback (AFF) are “semi-direct” effect, but with different cloud adjustments. For clarity, we have rephrased this statement to “The enhancement over ocean has been attributed to weakened cloud-top entrainment of overlying dry air due to absorbing aerosols from BB above the marine stratocumulus. The reduction of cloudiness over land can be explained by the aerosol absorption cloud fraction feedback (AFF) proposed by Koren *et al.*, where surface cooling reduces moisture fluxes and the

upper-level warming reduces the relative humidity in the cloud layer” (Lines 136-139).

Reference:

Hansen, J., Sato, M. & Ruedy, R. Radiative forcing and climate response. *J. Geophys. Res.* 102, 6831-6864 (1997).

Koch, D. & Del Genio, A. D. Black carbon semi-direct effects on cloud cover: review and synthesis. *Atmos. Chem. Phys.* 10, 7685-7696 (2010).

Ramanathan, V., Crutzen, P. J., Kiehl, J. T. & Rosenfeld, D. Aerosols, Climate, and the Hydrological Cycle. *Science* 294, 2119-2124 (2001).

Johnson, B. T., Shine, K. P., and Forster, P. M. The semi-direct aerosol effect: Impact of absorbing aerosols on marine stratocumulus, *Q. J. Roy. Meteorol. Soc.* 130, 1407–1422 (2004).

Wilcox, E. M. Stratocumulus cloud thickening beneath layers of absorbing smoke aerosol. *Atmos. Chem. Phys.* 10, 11769-11777 (2010).

Lu, Z. et al. Biomass smoke from southern Africa can significantly enhance the brightness of stratocumulus over the southeastern Atlantic Ocean. *Proc. Natl. Acad. Sci. U.S.A* 115, 2924-2929 (2018).

Furthermore, calling a process a feedback is typically reserved for a process where the response of the system to a forcing further affects the magnitude of the forcing. In what way is the proposed process over land a feedback? The reference provided for this “aerosol absorption cloud fraction feedback” is to a paper based on aerosol-cloud interactions over the Amazon, otherwise there is a long list of references purported to support the notion of a land/ocean contrast in the cloud response over southern Africa and the adjacent Atlantic Ocean. Is there a paper in the literature that provides evidence for the specific physical process leading to the reduction in cloud cover over land in Africa?

Response: As for the cloud reduction over land in Africa, there exists some literature to explore the underlying mechanism. Over the land in Africa, where the aerosols are often below or within cloud layers, the semi-direct effect adds to the positive radiative effect, making the impact of clouds on the total radiative effect at the TOA significantly positive. The reduction in cloud cover has been attributed to weaker convection driven by increased static stability due to BB aerosols (Sakaeda et al., 2011). On the contrary, in this work, we proposed the dominant role of BB smoke’s radiative effect on the low-

cloud enhancement over the land area of subtropical southeastern Asia. The main cause of the diverse response of cloud is different water vapor supply. In subtropical southeastern Asia located in the East Asian monsoon zone, low-level moisture convergence and wind from the South China Sea provide persistent and abundant water vapor supply for cloud formation, rather than predominantly from local surface evaporation.

The low-cloud enhancement over the land area of subtropical southeastern Asia can be termed as a feedback since the underlying processes do feature a positive loop. With intensive BB in Southeast Asia, the stratus cloud predominantly concentrates below 2 km and caps a moist boundary layer due to moisture convergence and wind from the South China Sea with abundant water vapor. Prevailing westerlies transport the BB smoke plume at an altitude around 3 km above the cloud layer, thereby producing a substantial atmospheric heating by light-absorbing BB aerosols. Then, a substantial free atmospheric warming tendency and a significant cooling in the PBL is formed downwind of the BB source region, particularly over the land area near 105 °E at the lee side of the plateau. Accordingly, more water vapor tends to condense into cloud droplets under a higher relative humidity in the dimming PBL, resulting in a substantial enhancement in cloud beneath the BB smoke plume in the downwind region. The thickened cloud further amplifies the PBL cooling and cloud formation under more moisture supply from the South China Sea by the weakened monsoon circulations (Fig. R3, shown as the revised Fig. 6, in which we added the “+” and “-” symbols to clarify the feedback loop). That is why we attribute the remarkable cloud enhancement to aerosol-cloud-PBL feedbacks. We have rephrased Lines 189-197 in the revision to better illustrate the aerosol-cloud-PBL feedbacks and the crucial role of monsoon system.

L189-197: Accordingly, more water vapor tends to condense onto cloud droplets under the higher relative humidity in the dimming region in the upper PBL, which are uplifted further by the converging circulations of the monsoon, and then transported eastward by the upper-level westerlies, resulting in a substantial enhancement in cloud beneath the BB smoke plume in the downwind region (Fig. 3c). It is worth noting that an increasingly thick and brighter cloud layer underneath the BB smoke plume further amplifies the PBL cooling and the heating tendency above, respectively, thereby reinforcing the positive aerosol-cloud-PBL feedbacks.

Fig. R3 | Mechanism of smoke-cloud-PBL feedback coupling with the monsoon in subtropical southeastern Asia. The gray shading indicates the biomass-burning plume. The red color in the plume shows shortwave heating by absorbing aerosols like BC above the clouds. Blue shading on the lee side of the Shan Plateau indicates the strong dimming caused by aerosol-cloud-PBL interactions over land. Yellow shading represents the water vapor supply. The “+” and “-” symbols indicate positive and negative feedback, respectively.

Reference:

Sakaeda, N., Wood, R. & Rasch, P. J. Direct and semidirect aerosol effects of southern African biomass burning aerosol. *J. Geophys. Res.-Atmos.* 116, D12205 (2011).

The argument for why the semi-direct effect may be stronger over subtropical Asia compared to the southeast Atlantic Ocean is attributed by the authors to stronger transport of low-level moisture into the cloud layer due to the monsoon circulation, which seems plausible. However, there is little discussion of the actual physics of the semi-direct effect in this region; or what the authors have labeled (but not really fully described) the “smoke-cloud-PBL feedback coupling”. The increase in cloud over the southeast Atlantic Ocean has been attributed in the past to a suppression of turbulent mixing of humidity from the cloudy boundary layer to the free troposphere above the cloud layer (Johnson et al 2004, Wilcox 2010). Are the authors arguing that the same process is occurring over subtropical Asia? If so, I think that the authors will have to consider some of the details of the vertical structure in the region: in particular the

relation of the cloud top height to the PBL height. Over the southeast Atlantic it is often the case (though not always) that the smoke is entirely, or at least predominantly, located in the free-troposphere and the clouds are capped by the strong subtropical inversion capping the marine boundary layer. This vertical structure is important for the proposed mechanism in that location because of the effect of stabilization of the lower free-troposphere inhibiting ventilation of the boundary layer through turbulent entrainment. None of the relevant physics of these processes or others that the authors might be arguing for over subtropical Asia is discussed.

Response: We greatly appreciated the reviewer’s suggestion on in-depth exploration of “smoke-cloud-PBL feedback coupling” and vertical structure of both aerosol and cloud. Regarding the difference in mechanisms between the two regions, vertically the semi-direct effect is similar, as the smoke plumes are generally above the cloud, while the unique part in Southeastern Asia is the role of the monsoon in the feedback loop and the role of PBL dimming over land PBL. Taking advice from both reviewers, we specifically investigated the impact of the aerosol-monsoon system and gained a more comprehensive understanding about the smoke-cloud-PBL-monsoon feedback.

We added the cross-section of PBL height in Fig. 3 of the revised manuscript (shown also as Fig. R4 below), which demonstrates the vertical structure of smoke, cloud (enhancement), and PBL. Also, we plotted vertical structures of wind and water vapor, as well as their changes that are induced by the aerosol-monsoon feedback in Fig. 5 (Fig. R5). We have also updated the conceptual scheme given in Fig. 6 to show the roles of main processes in the feedback loop (See Fig. R3) and added relevant discussions in the main text of the revision (Please refer to Lines 189-197, 261-273, and 279-281 in the revised main text).

Fig. R4 | WRF-Chem simulated low-cloud enhancement by semi-direct effect of biomass burning aerosols in subtropical southeastern Asia. **a**, Monthly averaged difference of low cloud fraction plotted with winds at an altitude of 1 km in March of high AOD years. **b**, Spatial patterns of BC concentrations and aerosol-induced atmospheric heating rate at the altitude of 3 km (red isolines with the unit of K day^{-1}) in March of high AOD years. **c**, Vertical cross-section of BC concentration, cloud enhancement and air temperature difference along the coastal region ($17^{\circ}\text{N} - 23^{\circ}\text{N}$) for the runs with BB on/off in March of high AOD years (2004, 2007, 2010 and 2014). **d**, **e**, **f**, Same as **a**, **b**, **c** but for low AOD years (2001, 2003, 2005, 2011). The results are calculated from the difference between the experiments EXP_FAR and EXP_exAR. Gray isolines in (**a**) and (**d**) show topography (Unit: km) and black dots mark the grids passing a T-test. Green lines in (**c**) and (**f**) show the boundary layer height at 6:00 UTC.

Fig. R5 | Synergetic effect of smoke-cloud-PBL feedback coupling with the monsoon circulation in subtropical southeastern Asia elucidated by WRF-Chem simulations. **a**, Monthly averaged column water vapor below 3 km and wind field at the altitude of 1 km in March at high AOD. **b**, Changes of column water vapor and pressure (blue contours, Unit: Pa) and wind field at the altitude of 1 km due to BB aerosols ARI effect. **c**, Vertical cross-section of water vapor with the wind field along the coastal region ($17^{\circ}\text{N} - 23^{\circ}\text{N}$) in (**a**). **d**, Same as (**c**) but for the changes due to BB aerosols ARI effect. Note: Results are averaged for March of high AOD years (2004, 2007, 2010, 2014). The BB aerosols ARI effect are shown by differences between the experiments EXP_FAR and EXP_exAR.

L189-197: Accordingly, more water vapor tends to condense onto cloud droplets under a higher relative humidity in the dimming region in the upper-PBL, which are uplifted further by the converging circulations of the monsoon and then transported eastward by the upper-level westerlies, resulting in a substantial enhancement in cloud (over 60%) beneath the BB smoke plume in the downwind region (Fig. 3c). It is worth noting that an increasingly thick and brighter cloud layer underneath the BB smoke plume further amplify the PBL cooling and the heating tendency above, respectively, thereby reinforcing the positive aerosol-cloud-PBL feedbacks.

L261-273: The aerosol-cloud-PBL feedbacks coupled with the unique monsoon regions could provide an interpretation on the difference compared to other regions. For example, over main biomass burning regions like Amazon, BB smoke has been proven to reduce cloudiness via suppressed moisture fluxes due to surface cooling and a lower relative humidity driven by aerosol heating, just as the originally termed semi-direct effect of aerosol. Similarly, the semi-direct effect of BB aerosol has been also postulated to thicken low cloud in southeast Atlantic. The moisture supply and PBL cooling take effect in oceanic and land areas separately (Supplementary Figs. 11 and 12), which thereby result in contrast effect on low cloud over the land and sea (Fig.1b). However, in subtropical Asia, the semi-direct effect of BB aerosol in the vertical direction greatly enhances low cloud as a horizontal abundant moisture supply under the influence of monsoon circulation feedback.

Line 279-281: Therefore, cloud cover enhancement in subtropical Asia is comparable to that over the southeast Atlantic even with much lower BB emissions, due mostly to the synergetic effect of aerosol-cloud-PBL-monsoon feedbacks.

The cross-sections through the simulated aerosol and cloud distributions in figures 3b and 3d are helpful for conveying the relation between the aerosols and clouds, but also raise many questions. First, the authors refer in line 147 to the “biomass burning source region”, but none of the details of the geography of the sources is discussed. Where exactly is the source region? What are the optical properties of the aerosols and are there published estimates of the magnitude of the radiative heating rate? Is the magnitude of the atmospheric temperature difference shown in figure 3b consistent with existing estimates of the shortwave radiative heating rate? All of these details are essential to arguing that the aerosols might be motivating a semi-direct effect and that the model simulations are representative of the realistic aerosol radiative heating in the atmosphere, yet none of this is discussed.

Response: Thanks for the suggestion on the detailed analysis of source region, optical properties, and heating tendency of BB aerosol. The source region of intensive BB activities predominately concentrates in the northern Indochina Peninsula, which is illustrated by the grey dots in Fig. 1. To make it clearer, we give the name of plateaus on the map of Supplementary Fig. 6 (i.e., Fig. R6).

For the purpose of evaluating the model's performance on capturing the optical properties of BB aerosol, we collected all the available measurements of aerosol absorption optical depth (AAOD) and relevant publications and conducted comparisons with the corresponding model simulations. As illustrated in Fig. R6 (also Supplementary Fig.6 in manuscript), the AAOD measurements at 31 AERONET stations are well reproduced by the WRF-Chem simulations, suggesting that the model is capable of describing the optical properties of the BB aerosol.

Fig. R6 | Model validation of simulated AAOD along the BB smoke plume. a, Locations of AERONET stations used to validate the simulated AAOD. Some AERONET stations are very close to distinguish from the map. **b,** Comparison between observed and simulated AAOD (EXP_FAR) at 400 nm wavelength at AERONET stations in March during 2001-2015. The whiskers show the standard deviation. The blue solid line gives the reduced major axis (RMA) regression for the scatters, and the fitted function is labelled in the top left corner.

Fig. R7 illustrates the distribution of biomass burning aerosols and the magnitude of the radiative heating rate (Fig. R7a) and temperature response (Fig. R7b) due to BB aerosols. It clearly demonstrates that the distribution of increased shortwave heating rate has a similar pattern as the spatial distribution of BC, suggesting that BB smoke containing a high concentration of light-absorbing BC substantially absorbs solar

radiation and heats the atmosphere (Fig. R7a). The temperature difference also has a similar distribution as the warming tendency above 1 K, indicating that the increased temperature is due to the enhanced shortwave heating rate. The average increased heating rate and temperature over the coastal region of southern China reached about 1~3 K/day and 1K, respectively. This result is consistent with Wilcox's (2010) study, which shows that the aerosols from biomass burning in Africa increase the heating rate by about 1.8 K/day and the monthly temperature about 1 K over the Atlantic. Furthermore, the heating rate is also supported by the study of Pani (2016), which shows the biomass burning aerosols' heating rate is about 1.4 K day⁻¹ in northern Indochina by using the 7-SEAS/BASELInE 2013 Campaign data.

Fig R7 | WRF-Chem Simulated BC and air temperature bias at 3-km altitude in March at high AOD. a, BC distribution at an altitude of 3 km with the aerosols' heating rate (show by contour line with unit: K/day). **b,** Same with (a) but for temperature bias due to BB aerosols ARI effect and aerosols' heating rate. Note: The dark circle in (b) shows the location of Nanning.

In addition to comparing model results with AERONET data and pre-existing studies, we further used radiosonde data to validate the model results and show how the biomass burning aerosols' radiation effect changes temperature stratification. At the Nanning station, which is strongly affected by biomass burning aerosols (location is shown in Fig R7b), radiosonde observations are carried out at 20:00 local time every day. Through comparing the vertical temperature profiles between observations and model results (Fig. R8), we can find that, by considering the biomass burning aerosols radiation effect, the simulated temperature vertical profile is more consistent with the observations, with the temperature increased by about 1-2 K around 700 hPa and decreased by about 5 °C near the surface. Considering the BC from biomass burning

mainly concentrates around 750 hPa, the temperature structure change suggests that the biomass burning aerosols heat the air around 700 hPa and cool the surface. A similar figure is in Extended Data Fig.9c in the previous version but with the BC concentration at 20:00 local time.

Accordingly, these details on the source region, optical properties and heating tendency of BB aerosol are added in the revised manuscript for clarity. Due to the length limitations, the validation of model results is provided in the Supplementary Data.

Fig R8 | Vertical distribution of air temperature from radiosonde measurements and different modeling scenarios for the eight highest aerosol-PBL interaction days at 20:00 local time in Nanning during March 2004 with the average BC concentration during daytime.

Reference:

Wilcox, E. M. Stratocumulus cloud thickening beneath layers of absorbing smoke aerosol. *Atmos. Chem. Phys.*, 10, 11769–11777 (2010).

Pani, S. K. et al. Radiative Effect of Springtime Biomass-Burning Aerosols over Northern Indochina during 7-SEAS/BASELInE 2013 Campaign. *Aerosol Air Qual. Res.* 16, 2802-2817 (2016).

Second, the smoke over the source region (shown in fig. 3b) is at the same elevation as

what I presume is the base of the cloud layer, or at least the lowest elevation where an enhancement of cloud is indicated to the east of the source region. For the semi-direct effect to be operating the smoke must be elevated above the cloud layer. The authors argue that monsoon convergence lofts the smoke above the cloud layer, but the only evidence presented to support this are small wind vectors in figures 3a and 3c. Perhaps there is literature that documents the link between the dynamics and the vertical profile of the smoke? Presumably mixing of the smoke with the cloud layer would also result in microphysical indirect effects that might be commingled with the semi-direct effects. Indeed, the cross section in extended data fig. 6a seems to imply that the smoke and clouds may be mixed at least as far east as 110 deg. E. Thus, I was struggling to reconcile the extinction cross-section in extended data fig. 6a suggesting strong coincidence of the smoke and cloud with the extinction profiles in fig. 2 which are presented to argue that the smoke is predominantly above the cloud. It is also unclear how then the authors can be sure that some of the effects they are seeing are not attributable to microphysical interactions of aerosols with clouds?

Response: Thanks for the comment. The mechanism of the elevation of smoke in this region has been studied in previous works (Lin et al., 2009; Jian et al., 2014; Ding et al., 2015, Xue et al., 2020). In general, the convergence of the surface wind field collects the smoke (Fig. R9a) and lifts it to an altitude of 3 km (Fig. R9b). In detail, the combination of the factors of convergence of wind field, orographic lifting, buoyancy uplifting and the lee-side troughs is responsible for the elevation of smoke. As the biomass burning mainly occurs in the highland with a terrain height of 1 km (Fig. R6a), the smoke from biomass burning can rise to a high level easily. On the other hand, the thermal energy of fire buoyancy the smoke and makes it easier to rise into the free troposphere (Jian & Fu, 2014). Besides, the lee-side trough also helps the smoke rise to a high level (Lin 2009; Ding et al., 2015).

Fig. R9 | Averaged BC concentration with wind field on the surface (a) and an altitude at 3 km (b) during March 2004.

Due to the mechanism mentioned above, the smoke is present mainly above the clouds, making the semi-effect happen. The conflicts that seem to exist between Fig. 2 and Extended Data Fig. 6a are caused by the low horizontal resolution of the CALIPSO monthly data, which has a resolution of 5 degrees in longitude and 2 degrees in latitude. At the same time, there is some mixing of clouds and aerosols. The mixing of smoke and clouds can be seen from CALIPSO profiles and MERRA2 data. Using the WRF-Chem model, we find that the microphysical interactions also increase the cloud fraction. However, the aerosols' microphysical effect on clouds is small compared to the radiation effect. This result has been shown in Supplementary Fig. 8 in the revised manuscript. The relatively small effects of aerosols and cloud microphysical interactions may be because anthropogenic aerosols have already provided high concentrations of CCN. Because of the pre-existing pollution, the additional ACI effect from the biomass burning aerosols will not be obvious (Twomey, 1984; Reutter, 2009).

Reference:

Ding, K., Liu, J., Ding, A., Liu, Q., Zhao, T.L., Shi, J., Han, Y., Wang, H., and Jiang F., Uplifting of carbon monoxide from biomass burning and anthropogenic sources to the free troposphere in East Asia, *Atmos. Chem. Phys.*, 15, 5, 2843-2866, 2015.

Lin, C. Y. et al. A new transport mechanism of biomass burning from Indochina as identified by modeling studies. *Atmos. Chem. Phys.* 9, 7901-7911 (2009).

Jian, Y. & Fu, T. M. Injection heights of springtime biomass-burning plumes over peninsular Southeast Asia and their impacts on long-range pollutant transport. *Atmos. Chem. Phys.* 14, 3977-3989 (2014).

Twomey, S. A., Pieprgrass, M. & Wolfe, T. L. An assessment of the impact of pollution on global cloud albedo. *Tellus B* 36, 356-366 (1984).

Reutter, P. et al. Aerosol- and updraft-limited regimes of cloud droplet formation: influence of particle number, size and hygroscopicity on the activation of cloud condensation nuclei (CCN). *Atmos. Chem. Phys.* 9, 7067-7080 (2009).

Xue, L., Ding, A., Cooper, O., Huang, X., Wang, W., Zhou, D., Wu, Z., McClure-Begley, A., Petropavlovskikh, I., Andreae, M.O., Fu, C.B., ENSO and Southeast Asian biomass burning modulate subtropical trans-Pacific ozone transport, *Nat. Sci. Rev.*, nwaal32, <https://doi.org/10.1093/nsr/nwaa132>, 2020.

The crux of the authors' argument is that the monsoon circulation acts to amplify the

semi-direct enhancement of cloud, as stated in the title. I found it rather difficult to follow the evidence for this. First, it is argued that the circulation “brings large amounts of water vapor ... [to] the lee side of the plateau”(line 178). The referenced figure in the extended data shows the circulation, but the transport of moisture is not presented quantitatively, so while this seems highly plausible, it is not demonstrated. I also note that the “plateau” is referenced a half-dozen times throughout the paper, but it is never made clear exactly which plateau is being referred to.

Response: Thanks for the comment. This comment and Reviewer #1’s first comment motivated us to further check the interaction between the aerosols’ semi-direct effect and the monsoon circulations based on WRF-Chem simulations. We found that the aerosol forcing did weaken the overall monsoon circulations and influenced the water vapor supply, promoting the accumulation of water vapor in the lower troposphere of the low-cloud enhancement region. We added the new Fig. 5 (shown also as Fig. R5) to quantitatively present the transport of moisture under the influence of aerosol-cloud-monsoon interactions. The new results make our previous discussion on the synergetic effect of aerosol-cloud-PBL-monsoon feedback more comprehensive and solid.

In the revised version, we added a paragraph to discuss these results. Also, we modified the conceptual scheme in Fig. 6 (Fig. R2) to consider these processes.

For the terminology of the plateau, we have added one map in Supplementary Fig. 6a (see also Fig. R6a) to define all the geographic terms and changed the relevant discussions in the main text for clarity.

Next it is argued that the “air masses become saturated in the middle and upper PBL by ARI-induced cooling” (line 180). But the figures referenced for this indicate cooling only near the surface and apparently below the level of enhanced cloudiness shown in figure 3b, so it was not clear to me from the evidence provided that the cooling is truly where the clouds are being enhanced. Also, since there is no discussion of the PBL structure, or its relation to the aerosol and clouds (as noted above) it is hard to know exactly what the authors mean when they say “middle and upper PBL”.

Response: The BB smoke aloft led to a substantial free atmospheric heating and PBL cooling on the lee side of the plateau. With the abundant water vapor supply by the monsoon, air masses become saturated in the middle and upper dimming PBL by ARI-induced cooling. The air temperature stratification response to ARI is characterized by cooling below 1.0 km as well as warming above 2.0 km (Fig. 3c, i.e., Fig. R4c), which

covers the “the level of enhanced cloudiness shown in Fig. 3c”. The original Fig. 3 shows the substantial relative increase of cloud fraction by the ARI effect of BB smoke. In terms of the changes in cloud liquid water content, the most obvious increase in cloud liquid water content (>30 mg/kg) is concentrated in the dimming region below 1.0 km (Fig. 3c). This enhanced low cloud liquid water content is then uplifted by the converging circulation and transported eastward by the upper-level westerlies, resulting in a substantial enhancement in cloud around 1.5 km. Also, the warming tendency above the low cloud greatly weakens the turbulent motion. Thus, it then causes a reduction in cloud-top entrainment of dry air, leading to further thickening of the cloud (Deardorff, 1980; Mellado, 2017). Regarding to the term of “middle and upper PBL”, we have added the PBL height in the newly-added Fig. 3c, and rephrased “middle and upper PBL” to the specific altitudes.

Reference:

Deardorff, J. W. Cloud top entrainment instability. *J. Atmos. Sci.* 37, 131-147 (1980).

Mellado, J. P. Cloud-Top Entrainment in Stratocumulus Clouds. *Annu. Rev. Fluid Mech.*, 49, 145-169 (2017).

Next it is argued that “radiative heating of smoke above clouds creates a strong inversion that is conducive to the formation of extensive shallow clouds below it” (line 183). But the only figure referenced for this is a cartoon graphic (fig. 4). There are some temperature profiles presented in the extended figures, but there is no discussion of any details regarding how the aerosol heating is impacting the vertical temperature structure, or where the inversion is expected to be relative to the aerosols and clouds in this region, or the magnitude of any strengthening of the inversion. Furthermore, there is no discussion of the relevant physics by which one might expect an enhanced inversion to promote shallow cloud development.

Response: Thanks for the comment. In the revised manuscript, we have added the monthly profile of temperature responses to the radiative effect of smoke aerosol above the clouds (Fig. 3c) and the simulated diurnal variations of temperature changes (Fig. 4c). Both indicate a strong inversion due to BB smoke, which is characterized by cooling below 1 km as well as warming around the altitude of 2-3 km. We also compared the temperature profile with/without ARI by BB smoke in Fig. 4a and Supplementary Fig. 9 in the revised manuscript, which is also indicative of an obvious temperature inversion around 750 hPa, that is around 2-3 km. To be clearer, Fig. R10

shows temperature changes due to BB aerosols. It can be seen that the BB aerosols heat the air around an altitude of 3 km and cool the surface. The most significant temperature changes are on the east side of the Indochina Peninsula, where the cloud fraction is increased obviously (Fig. R3a). On one hand, the near-surface cooling makes the air mass readily reach saturation, thus increasing the low clouds. On the other hand, the warming tendency around 3 km stabilizes the low atmosphere and reduces the turbulent mixing of humidity above the cloud layer (Johnson, 2004; Wilcox, 2010; Deardorff, 1980; Mellado, 2017), further thickening the low clouds beneath. The enhanced cloud would then reflect more solar radiation, further cool the ground surface and lead to more light absorption by BB smoke, resulting in a more stabilized low troposphere and cause positive feedback between BB aerosol, PBL and clouds.

Fig R10 | (a) Changes of temperature and aerosols' heating rate (show by contour line with unit: K day⁻¹) at an altitude of 3 km and (b) on the surface due to biomass burning aerosols radiation effect.

Accordingly, we add the spatial distribution of temperature difference in Supplementary Fig. 13 in the revised manuscript.

Reference:

Johnson, B. T., Shine, K. P., and Forster, P. M.: The semi-direct aerosol effect: Impact of absorbing aerosols on marine stratocumulus, *Q. J. Roy. Meteorol. Soc.* 130, 1407–1422 (2004).

Wilcox, E. M.: Stratocumulus cloud thickening beneath layers of absorbing smoke aerosol. *Atmos. Chem. Phys.* 10, 11769–11777 (2010).

Deardorff, J. W. Cloud top entrainment instability. *J. Atmos. Sci.* 37, 131-147 (1980).

Mellado, J. P. Cloud-Top Entrainment in Stratocumulus Clouds. *Annu. Rev. Fluid Mech.* 49, 145-169 (2017).

Finally, there is a single mention of the shallow clouds being “maintained by the radiative cooling over land associated with the transport of water vapor by the monsoon circulation” (line 185), again referencing the cartoon rather than a paper (of which there are presumably many in the monsoon literature) or a quantitative argument based on the authors’ analysis or modeling. This link to the circulation is what I presume the authors mean when they say that the Asian monsoon “amplifies” semi-direct effects in the paper’s title. In the discussion it is not presented as an “amplification”, rather it is described as a “synergetic effect of aerosol-cloud-PBL feedbacks and the large-scale monsoon” (line 190). But presumably they mean that the aerosol-cloud-PBL dynamics feeds back ON the monsoon circulation? It is not really clear which elements of this system are argued to be acting as feedbacks, or exactly how these elements are acting synergistically.

Response: Thanks for the comment. According to your suggestion, we further checked the monsoon circulation response to the BB aerosol. A warming core at 3 km triggers an abnormal low-pressure cyclonic circulation along the coastal region of southern China below the smoke (Fig. 5b in the revised manuscript, also see Fig. R5b), where the moisture convergence is intensified in the region of low-cloud enhancement. We added Fig. 5 to discuss the aerosol impact on the monsoon circulation and its further impact on the water vapor supply quantitatively.

Regarding to the comments on the title, we changed it into “Aerosol-PBL-monsoon interactions amplify semi-direct effect of biomass smoke on low cloud formation in Southeast Asia”, which is better suited to highlighting our overall findings on the “synergetic effect of aerosol-cloud-PBL feedbacks and the large-scale monsoon”. We believe that the new-added results do show the “*amplification*” effect in the feedbacks (as that shown in Fig. 6).

I get a sense that there is the nugget of an interesting idea in this paper, but the evidence is simply not presented in a quantitatively convincing manner. Furthermore, the language throughout the manuscript, from the geography, to the physics, to buzzwords like “synergy” and “feedback”, are simply not used precisely enough to make clear what exactly is being argued to be occurring. Part of the problem may be

that the overall structure of the paper is ill suited to a letter journal owing to the fact that the authors have tried to present results from multiple regions with differing responses and then build their argument for why the subtropical Asian monsoon region is different. Although the structure of the argument is compelling, the authors have included a large number of figures to compare and contrast the detailed differences in the arrangement of aerosols and clouds in the observations in multiple locations and the model simulations of the monsoon system. The many references to both extended data and supplementary content begs the questions of why a letter journal was considered the right choice for a paper requiring so many figures but also subjected to such a limiting maximum word count. In many cases there are declarative statements with references to multiple figures, but little description of what the figure depicts or explanation of the relevant physics at work. One strategy to improve upon this could be to simply use the existing literature to back up claims about semi-direct effects in regions outside of subtropical Asia and then focus the discussion on describing in better detail the authors' arguments for the nature of semi-direct effects in subtropical Asia and their relationship to the monsoon circulation. Choosing a journal that allows more words and figures would perhaps make that easier to accomplish.

Response: Thanks for the comments. Our previous version was transferred from Nature as a "letter". The Nature Communications Article allows more figures and discussions. In the revised manuscript, we extended the number of figures and added more discussions on the key conclusions. Also, we removed some unnecessary figures to make the overall story clearer. Thanks again for your advice.

Additional recommendation:

The authors claim on line 88 and 89 that the emissions of biomass burning in Asia are 30% of that in southern Africa; a point that was also mentioned in the abstract. The implication seems to be that the semi-direct effect is somehow more efficient, per unit of emissions in Asia, although the authors do not explicitly make that point. This result is based on emissions estimates in units of mass of carbon per unit area. From a land-use perspective, perhaps it might be meaningful to relate emissions to an aerosol-cloud effect in this way, however, the semi-direct effect responds to the radiative forcing of the aerosol, not its emissions rate. There are many other factors that could lead to different radiative forcing for the same rate of emissions, including the radiative properties of the particles, their residence time in the atmosphere, the concentration of

particles in the resulting atmospheric plume, etc... This point is only incidental to the main goal of the paper, however, it is included in the abstract of the paper. If the authors are going to emphasize this point, then they should clarify exactly what the reader should be taking away from this result and why emissions rate is the right quantity to compare between the two regions. If indeed the point is to suggest that the cloud enhancement is more effective over subtropical Asia, then I think they need to make a more quantitative case to back that up.

Response: Thanks for the recommendation. We agree that here we shouldn't use the emission rate but instead of the total emission. Comparison with Africa, where the impact of biomass burning is stronger and the mechanism has been well-studied, is one of the key points that we would like to highlight in this manuscript. As shown in Supplementary Fig. 2, even though the carbon emission of biomass burning in the Indochina Peninsula is only 30% of that in Africa, the enhanced cloud cover area in subtropical Asia resulting from the aerosols' effect is almost the same, from the perspective of area and intensity.

In the revision, we updated the emission rate into total carbon emission in Supplementary Table 1 and changed relevant discussions in the main text and abstract. As shown, the total emission in Asia is only about 20% of that in Africa.

REVIEWER COMMENTS

Reviewer #1 (Remarks to the Author):

I would like to thank the authors for the comprehensive reply to my comments and for the additional analysis. My concerns have been fully addressed in the revisions manuscript. I am happy to recommend the paper for publication.

Reviewer #2 (Remarks to the Author):

I still feel that this paper presents an intriguing hypothesis. This version of the paper is an improvement over the prior submission, in particular regarding the aspects of the argument related to the role of the monsoon circulation. Nevertheless, the imprecise use of language in the narrative leaves me unconvinced, and I still cannot support publication of the paper in this form. There are numerous references to "feedbacks" (or sometimes just a "feedback") that are either not demonstrated to be at work, or in some cases not referring to an actual feedback process at all. There is reference to a "synergy" that is not quantitatively justified. All of this seems to obscure the description of the physics at work in the authors model simulations.

The important improvement in this version of the paper over the previously submitted version is clearer exposition and better justification for the role of the large-scale monsoonal circulation in the overall response of the simulated system to the aerosol radiative impacts. The addition of figure 5 better demonstrates how the changes in the monsoon attributed to the aerosol effect in the WRF-Chem simulations impacts the water vapor over the ocean and coastal continental regions.

In the authors response to the concern of reviewer 1 about whether or not the ECMWF reanalysis includes the aerosol effects, the authors note only that departures from the model prior exceeding a threshold (which is noted as 5 standard deviations in the authors' response, but not in the revised manuscript) are not assimilated. The authors then conclude that "therefore the OMR approach can shed light on the aerosol effects that haven't been included" (lines 76-78). However, it is not obvious that the aerosol effects necessarily exceed that threshold. Five papers are cited after the sentence. Perhaps the quantitative evidence is in one of these papers. Do I need to read 5 papers to figure out which one includes the quantitative verification of that? This seems almost like a slight-of-hand. I know it is probably not, and as the authors argue in their response to the reviewer, many other papers have used this approach to diagnose aerosol radiative effects. So why not plainly state that cases of high aerosol heating typically exceed the threshold causing them to be excluded from the assimilation, and then cite the one paper that demonstrates that quantitatively?

I still find the multitude of references to "feedbacks" confusing and unsubstantiated. Here are a few separate, but related comments on this:

(a) It seems the most frequent invocation of a feedback is the aerosol-cloud-boundary-layer feedback noted in the abstract. Presumably, this is the feedback described by the authors in lines 198-200, where they state: "It is worth noting that an increasingly thick and brighter cloud layer underneath the BB smoke plume further amplifies the PBL cooling and the heating tendency above, respectively, thereby reinforcing the positive aerosol-cloud-PBL feedbacks." Even the use of the word "feedback" is rather recursively applied here. If aerosol cooling of the PBL is enhancing cloud that further cools the PBL, then that would be the positive feedback process rather than a process that is "reinforcing the positive ... feedbacks". Regardless, I cannot really see any clear evidence from the paper that this positive feedback is actually occurring and contributing to

greater cloud cover. While certainly a plausible hypothesis, it seems that one could equally conclude that the adjustment of the monsoon circulation might be enough to cause in the increase in cloud, could they not?

(b) The caption of figure 4 seems to conflate the process of aerosol radiative interactions with an "aerosol feedback". A model experiment that compares a simulation of aerosols that interact with radiation to a simulation of aerosols that do not interact with radiation is not the same as comparing a simulation with "aerosol feedback" to one without "aerosol feedback". In this context, it is not even really clear what is meant by "aerosol feedback". The aerosol radiative interactions act as a forcing on the dynamics of the model. There could perhaps be some feedbacks in the complex dynamical response to that forcing. But the radiative interactions themselves are not "feedbacks". Note that this conflation of aerosol radiative interactions and "aerosol feedback" is also present in the methods section of the paper.

(c) The paragraph beginning on line 266 begins by referring to the aerosol-cloud-PBL feedbacks discussed above. Then on line 278 refers to a new, previously unmentioned "monsoon circulation feedback". Then the paragraph finishes by referring to the synergy of some previously unmentioned "aerosol-cloud-PBL-monsoon feedbacks". I think the authors have identified a quite interesting regional circulation response in WRF-Chem to aerosol radiative forcing in the Asian Monsoon region that promotes low cloud development. However, it is not demonstrated that specific feedback processes are at work, or are even necessary for the enhancement of low cloud development the authors are arguing accompanies the biomass burning aerosols. Again, can the authors demonstrate that an adjustment of the monsoon circulation alone is sufficient to explain the changes in cloud? The many frequent references to feedbacks (or sometimes just a singular feedback), as well as the suggestion of an unquantified "synergy" imply a complex set of processes that are not clearly demonstrated.

Response to reviewer: 2

I still feel that this paper presents an intriguing hypothesis. This version of the paper is an improvement over the prior submission, in particular regarding the aspects of the argument related to the role of the monsoon circulation. Nevertheless, the imprecise use of language in the narrative leaves me unconvinced, and I still cannot support publication of the paper in this form. There are numerous references to “feedbacks” (or sometimes just a “feedback”) that are either not demonstrated to be at work, or in some cases not referring to an actual feedback process at all. There is reference to a “synergy” that is not quantitatively justified. All of this seems to obscure the description of the physics at work in the authors model simulations.

Response: Thank you for raising these comments regarding to the feedback(s). As shown in the conceptual scheme of Fig. 6, there is indeed a feedback loop contributed to the substantial enhancement of low clouds, and the role of most of the processes have been demonstrated by our observational/modeling evidence. We agree that the role of the key processes could be more quantitatively represented. Since the smoke-cloud-PBL interaction in the vertical direction and the adjustment of monsoon circulation in the horizontal direction are the two key processes in the feedback loop, we carried out additional numerical modeling experiments to quantitatively show their roles. For the smoke-cloud-PBL interaction, we conducted 1-D WRF-Chem simulations and compared the results from experiments with/without cloud processes to demonstrate the role of low cloud formation in the feedback. We carried out 3-D WRF-Chem simulations by assimilating the wind field from aerosol-meteorology fully-coupled modeling to quantify the role of the adjusted monsoon circulation in enhancing the low cloud. (Please see the response to the comments (a) & (c) and Fig. R2 and R3 for details).

We believe that with these additional modeling results, the “synergetic feedback” from the horizontal and vertical processes is now more clearly presented in a quantitative manner.

The important improvement in this version of the paper over the previously submitted version is clearer exposition and better justification for the role of the large-scale monsoonal circulation in the overall response of the simulated system to the aerosol radiative impacts. The addition of figure 5 better demonstrates how the changes in the monsoon attributed to the aerosol effect in the WRF-Chem simulations impacts the water vapor over the ocean and coastal continental regions.

Response: Many thanks for the overall encouraging comments. We do appreciate the suggestions that raised during the first-round review.

In the authors response to the concern of reviewer 1 about whether or not the ECMWF reanalysis includes the aerosol effects, the authors note only that departures from the model prior exceeding a threshold (which is noted as 5 standard deviations in the authors’ response, but not in the revised manuscript) are not assimilated. The authors then conclude that “therefore the OMR approach can shed light on the aerosol effects that haven’t been included” (lines 76-78). However, it is not obvious that the aerosol effects necessarily exceed that threshold. Five papers are cited after the sentence.

Perhaps the quantitative evidence is in one of these papers. Do I need to read 5 papers to figure out which one includes the quantitative verification of that? This seems almost like a slight-of-hand. I know it is probably not, and as the authors argue in their response to the reviewer, many other papers have used this approach to diagnose aerosol radiative effects. So why not plainly state that cases of high aerosol heating typically exceed the threshold causing them to be excluded from the assimilation, and then cite the one paper that demonstrates that quantitatively?

Response: Thank you for raising this point. Because the detailed information on ECMWF assimilation, i.e., which data have been excluded during the operational run of the model, is not publicly available, in our previous response to the Reviewer #1, we tried to clarify the validity of the OMR method based on relevant works using similar methods (e.g., Kalnay et al. 2003; Zhao et al., 2014), observational evidence during haze events in China (Ding et al., 2013; Huang et al., 2018), and general information about the methodology applied in the ECMWF data assimilation system (Tavolato et al., 2010; Dee et al., 2011). Unfortunately, so far none of these studies plainly stated that “the cases of high aerosol heating existing the threshold causing them to be excluded from the assimilation”. However, the misrepresentation of the surface temperature in the ECMWF during the fire events is clearly shown by the up to 10°C difference in surface air temperature between ECMWF and observation (Fig. 4a; Supplementary Fig. 9).

Based on multi-year GFS, GDAS and radiosonde data, our recent study (Huang and Ding, 2021) provides clear global evidence that the real-time aerosol effect hasn't been adequately represented in current weather prediction models and the reanalysis with data assimilation. As shown in Fig. R1, strong OMR and OMF (observation minus forecast) values existed in regions of high aerosol loading and the assimilated modeling output GDAS can only catch about half of the observed air temperature bias (Fig. R1c, slope=2.1). This means that many observation data with higher differences (i.e., over a certain number of standard deviations) have been excluded in the data assimilation. In the revised manuscript, we use Huang and Ding (2021) as the main reference to demonstrate the validity of the OMR method.

As we mentioned in our previous response, the OMR method was used to introduce the overall story and to identify the hotspots. Based on the existing literature and observational evidence in Asia, we believe that the OMR method does reveal the potential aerosol effect, which is also consistent with our modeling results demonstrated in this study. However, we do appreciate the suggestion by the reviewer. It could be a potentially interesting topic to be further investigated on how many observations in polluted region are excluded in the ECMWF assimilation in the future, of course, through collaborations with staff at the ECMWF.

Fig. R1 Air temperature forecast bias from OMF (Observation minus Forecast) and OMR (Observation minus Reanalysis) for GFS data during 2016-2018. (a) 925 hPa observation minus forecast (OMF) air temperature bias (T_{bias}) for the GFS 24-h forecast compared with the GDAS analysis (GDAS-GFS, shaded contour) and radiosonde observations (radiosonde-GFS, circles) during 2016–2018. The global mean forecast bias (GM) is labeled on the color bar. (b) The global distribution of the averaged MISR aerosol optical depth (AOD) during 2016–2018. (c) Scatter plot of OMF 24-hour air temperature bias, T_{bias} , at 925 hPa compared with that from radiosonde observations (Radiosonde-GFS) and from corresponding analysis data (GDAS-GFS), respectively during 2016-2018. Data locations are shown in Fig. R1a. (d) Vertical profile of 24-hour air temperature forecast bias compared with radiosonde observations over land when $\text{AOD} \times \text{SW}$ is less than 1.0. (e) Vertical profile of 24-hour temperature forecast bias compared with radiosonde observations over land when $\text{AOD} \times \text{SW}$ greater than 1.0. Green and orange lines present biases from radiosonde observations and corresponding analysis data, respectively. Shadows and bars represent 25-75th percentile ranges. (Results from *Huang and Ding, 2021@Science Bulletin*)

Reference:

Huang, X., and Ding, A., Aerosol as a critical factor causing forecast biases of air temperature in global numerical weather prediction models, *Science Bulletin*, <https://doi.org/10.1016/j.scib.2021.05.009> (2021).

I still find the multitude of references to “feedbacks” confusing and unsubstantiated. Here are a few separate, but related comments on this:

- (a) It seems the most frequent invocation of a feedback is the aerosol-cloud-boundary-layer feedback noted in the abstract. Presumably, this is the feedback described by the authors in lines 198-200, where they state: “It is worth noting that an increasingly thick and brighter cloud layer underneath the BB smoke plume further amplifies the PBL cooling and the heating tendency above, respectively, thereby reinforcing the positive aerosol-cloud-PBL feedbacks.” Even the

use of the word “feedback” is rather recursively applied here. If aerosol cooling of the PBL is enhancing cloud that further cools the PBL, then that would be the positive feedback process rather than a process that is “reinforcing the positive ... feedbacks”. Regardless, I cannot really see any clear evidence from the paper that this positive feedback is actually occurring and contributing to greater cloud cover. While certainly a plausible hypothesis, it seems that one could equally conclude that the adjustment of the monsoon circulation might be enough to cause in the increase in cloud, could they not?

Response: Thank you for raising these comments regarding to the “feedback(s)”. Here, we conducted two additional simulations to support our discussion on the aerosol-cloud-PBL interaction in the vertical direction and to quantify the role of the adjusted monsoon circulation in the horizontal direction.

In order to confirm the aerosol-cloud-PBL interaction, we carried out 1-D simulations using the single column model of WRF-Chem (Wang et al., 2018) for the case shown in Fig. 4ab (at Wuzhou on 12 March 2004) with a particular focus on the role of cloud. As summarized in Table R1, three experiments have been conducted: (1) CEXP_exAR (without the effect of aerosols on radiation), (2) CEXP_AR&exCR (with the effect of aerosols on radiation but without that of clouds on radiation), and (3) CEXP_AR&CR (with both effects of aerosols and clouds on radiation). As that show in Fig. R2 (i.e., the Supplementary Fig. 10 in the revised manuscript), low clouds play an important role in the feedback and themselves were enhanced in the upper-PBL because of the increased relative humidity associated with the entire PBL dimming. The smoke heating above the cloud is also substantially enhanced. These results, together with the 3-D simulation presented in Fig. 3 and Fig. 4, confirm the importance of the “aerosol-cloud-PBL feedback” (replaced by “aerosol-cloud-PBL interaction in the revision) in the vertical direction. We added these results in the Supplementary Materials (Supplementary Fig. 10) and discussed them in the main text in Lines 227-229 and Methods section in Lines 404-418.

Table. R1. WRF-Chem column model parallel numerical experiment designs.

Experiment	Aerosols impact on radiation	Clouds impact on radiation
CEXP_exAR	×	√
CEXP_AR&exCR	√	×
CEXP_AR&CR	√	√

Lines 228-230: This kind of aerosol-cloud-PBL interaction is also demonstrated by a case study for 13 March 2004 at Wuzhou using a 1-D WRF-Chem simulation (Methods and Supplementary Fig. 10).

Lines 405-422: To demonstrate the aerosol-cloud-PBL interaction response to low-cloud enhancement, particularly the role of upper PBL clouds, we conducted additional 1-D (single column) WRF-Chem simulations at Wuzhou on 13 March 2004 (i.e., the case shown in Fig. 4ab). The simulations were initiated with meteorological profiles over Wuzhou from the EXP_ARI simulations at 06:00 local time on 12 March (i.e., with an 18-hour spin-up time). For the initial condition of the aerosol profile, the maximum profile of 13 March was used to keep the 1-D

simulation similar to the overall aerosol profile on 13 March over Wuzhou. Three experiments were conducted: *CEXP_exAR* (without the effect of aerosols on radiation), *CEXP_AR&exCR* (with the effect of aerosols on radiation but without that of clouds on radiation) and *CEXP_AR&CR* (with the effects of both aerosols and clouds on radiation) (See Supplementary Table 4). As shown in Supplementary Fig. 10, low clouds play an important role in the interaction and were themselves enhanced in the upper-PBL (Supplementary Fig. 10a) because of the increased relative humidity associated with the entire PBL dimming (Supplementary Fig. 10b). The smoke heating above the cloud is also substantially enhanced (Supplementary Fig. 10b). These results, together with the 3-D simulation presented in Fig. 3 and Fig. 4, confirm the importance of aerosol-cloud-PBL interactions in the vertical direction.

Fig. R2. The role of aerosol-cloud-PBL interactions demonstrated by 1-D WRF-Chem simulations for 13 March 2004 at Wuzhou. (a) Changes in air temperature and clouds due to aerosol-cloud-PBL interaction. (unit: K day^{-1}). The red lines and blue lines indicate increase and decrease of air temperature. Shaded gray and color contours show BC concentration and change in cloud water. **(b)** Changes in relative humidity and short-wave heating due to the influence of aerosols and clouds. The blue contour lines show the increased cloud water (unit: g kg^{-1}) and the red lines show the enhanced short-wave heating rate of aerosols by the aerosol-cloud-PBL interactions (unit: K day^{-1}). Note: The changes in temperature, cloud water and relative humidity are calculated from the difference between the experiments *CEXP_AR&CR* and *CEXP_exAR*. The enhanced short-wave heating rate is calculated from the difference between the experiments *CEXP_AR&CR* and *CEXP_AR&exCR*

Regarding the role of the adjusted monsoon circulation, our main argument on its potential role in the “synergetic feedback” is that it provides more water vapor in the upper PBL (as shown in Fig. 5d). However, because this kind of monsoon change is a combined effect forced by the large-scale aerosol-cloud-radiation interaction in the entire region, it is very difficult to separate it out from the simulation with multi-processes. To overcome this challenge, we designed the following experiments using WRF-Chem: *EXP_ARIwind_exARITemp_ndg*, where the wind field is nudged to that of the simulation with ARI (i.e., to mimic the wind circulation of the adjusted monsoon in *EXP_ARI*), and air temperature is nudged to that without ARI effect (i.e., no influence from aerosol-cloud-radiation interaction, *EXP_exAR*). The difference between experiment *EXP_ARIwind_exARITemp_ndg* and *EXP_exAR* could then give a quantitative estimation of the role of the adjusted monsoon circulation in the synergetic feedback.

As shown in Fig. R3, the adjusted monsoon circulation did play a role in this kind of feedback to enhance the low-cloud below the smoke plume. However, a comparison of Figs. R3a and R3b suggest that the adjusted monsoon contributed about 25% of the enhanced low cloud, with the rest contributed mainly by the aerosol-cloud-PBL interaction. Note here, the adjustment of the monsoon itself is caused by the aerosol-radiation effect as shown in Fig. 5. These results give a clearer picture about the role of different processes in the synergetic feedback. We added these results in the Supplementary Materials (Supplementary Fig. 14) and discussed them in the main text in Lines 287-297 and Methods section in Lines 419-426

Fig. R3. Increased cloud cover due to synergetic feedback and adjusted monsoon circulation alone in high AOD years (2004, 2007, 2010 and 2014). (a) Increased cloud covers due to the ARI effect of BB smoke aerosol (EXP_ARI - EXP_exAR). (b) Increased cloud covers due to the adjusted monsoon circulation alone (EXP_ARIwind_exARITemp_ndg - EXP_exAR). Note: EXP_ARI - with aerosol-radiation interaction, EXP_exAR - without aerosol-radiation interaction, EXP_ARIwind_exARITemp_ndg - with wind nudged to that of the simulation with ARI (i.e., the adjusted monsoon circulation) and air temperature nudged to that without ARI effect (i.e., no influence from aerosol-cloud-radiation interaction).

Lines 287-298: Although the adjusted monsoon circulation was also associated with the aerosol-cloud-PBL interaction at the regional scale, one might raise the question if the adjusted monsoon alone could cause this kind of cloud enhancement. To quantify the role of the monsoon change, we

conducted another WRF-Chem experiment (*EXP_ARIwind_exARITemp_ndg*, *Methods*), in which the wind was nudged to that of the simulation with ARI effect (*EXP_ARI*) and the air temperature was nudged to that without ARI effect (*EXP_exAR*). The difference between *EXP_ARIwind_exARITemp_ndg* and *EXP_exAR* can give a quantitative estimation of the role of the adjusted monsoon circulation in the synergetic feedback. A comparison of Supplementary Figs. 14a and 14b suggests that the adjusted monsoon contributed about 25% of the enhanced low cloud, with the rest contributed mainly by the aerosol-cloud-PBL interaction.

Lines 423-430: We further conducted another WRF-Chem experiment to quantify the influence of the adjusted monsoon on the synergetic feedback (EXP_ARIwind_exARITemp_ndg), in which the wind was nudged to that of the EXP_ARI simulation (i.e., the adjusted monsoon circulation) and the air temperature was nudged to that of the EXP_exAR simulation (i.e., no influence from aerosols). The difference between EXP_ARIwind_exARITemp_ndg and EXP_exAR can give a quantitative estimation of the role of the adjusted monsoon circulation in the synergetic feedback (Supplementary Fig. 14).

In addition, we agree that the term “feedback” in different contexts may cause some confusion. So, we changed it into “interaction” in many places, but for the “synergy” effect we would like to retain the word “feedback” because the additional quantitative results together with existing observational and modeling results provide quite solid evidence. In fact, the positive feedback loop is the key to explain why the low-cloud enhancement are particularly strong in southeast Asia.

Reference:

Wang, Z., Huang, X, and Ding, A., Dome effect of black carbon and its key influencing factors: a one-dimensional modelling study, *Atmos. Chem. Phys.*, 18, 4, 2821, 2834, 2018.

(b) The caption of figure 4 seems to conflate the process of aerosol radiative interactions with an “aerosol feedback”. A model experiment that compares a simulation of aerosols that interact with radiation to a simulation of aerosols that do not interact with radiation is not the same as comparing a simulation with “aerosol feedback” to one without “aerosol feedback”. In this context, it is not even really clear what is meant by “aerosol feedback”. The aerosol radiative interactions act as a forcing on the dynamics of the model. There could perhaps be some feedbacks in the complex dynamical response to that forcing. But the radiative interactions themselves are not “feedbacks”. Note that this conflation of aerosol radiative interactions and “aerosol feedback” is also present in the methods section of the paper.

Response: Thanks for the comments. We have checked the main text and methods section of the paper throughout and replaced it with more precise words. For example, we used “aerosol-radiation interaction (ARI)”, “ARI effect” and “Aerosol-cloud-PBL interaction”. We only keep using “feedback” in the discussion on the synergetic feedback loop (as that demonstrated in Fig. 5).

(c) The paragraph beginning on line 266 begins by referring to the aerosol-cloud-PBL feedbacks discussed above. Then on line 278 refers to a new, previously unmentioned “monsoon circulation feedback”. Then the paragraph finishes by referring to the synergy of some

previously unmentioned “aerosol-cloud-PBL-monsoon feedbacks”. I think the authors have identified a quite interesting regional circulation response in WRF-Chem to aerosol radiative forcing in the Asian Monsoon region that promotes low cloud development. However, it is not demonstrated that specific feedback processes are at work, or are even necessary for the enhancement of low cloud development the authors are arguing accompanies the biomass burning aerosols. Again, can the authors demonstrate that an adjustment of the monsoon circulation alone is sufficient to explain the changes in cloud? The many frequent references to feedbacks (or sometimes just a singular feedback), as well as the suggestion of an unquantified “synergy” imply a complex set of processes that are not clearly demonstrated.

Response: Thanks for the comments and suggestions. For the wording of “feedback”, we changed it into “aerosol-cloud-PBL interaction” and only use it in the discussions of the synergy feedback loop, i.e., the combined effect of aerosol-cloud-PBL interaction and the adjustment of monsoon circulation. As shown in Fig. R2, clouds in the upper PBL play important role in the aerosol-cloud-PBL interaction, and the adjusted monsoon circulation contributed mainly to the water vapor transport and accumulation, which may further promote cloud formation. The adjustment of circulation alone contributed about 25% of the low cloud enhancement (Fig. R3).

REVIEWERS' COMMENTS

Reviewer #2 (Remarks to the Author):

I appreciate the authors substantial efforts to address the comments I have made. Although it may seem like nit-picking, I do sincerely believe that the more precise use of language has allowed the revised manuscript to more clearly describe the physical processes that authors are arguing are occurring. The additional simulations the authors have performed have also gone a long way to strengthening their arguments. The paper in its present form is suitable for publication in my estimation. I commend the authors for a very comprehensive and insightful study.